# Angiogenic and immune predictors of neoadjuvant axitinib response in renal cell carcinoma with venous tumour thrombus

Rebecca Wray [1,2,3,32], Hania Paverd [1,2,3,4,32], Ines Machado [1,3], Johanna Barbieri[5], Farhana Easita[3], Abigail R. Edwards [5], Ferdia A. Gallagher[4,6], Iosif A. Mendichovszky[4,6], Thomas J. Mitchell [1,4,7], Maike de la Roche [5], Jacqueline D. Shields [8], Stephan Ursprung[4,6], Lauren Wallis[9], Anne Y. Warren [4], Sarah J. Welsh[10], Mireia Crispin-Ortuzar [1,2,3,33], Grant D. Stewart [3,4,7,33] & James O. Jones[2,3,4,5,33] ✉ On behalf of the NAXIVA Study Group*

Venous tumour thrombus (VTT), where the primary tumour invades the renal vein and inferior vena cava, affects 10–15% of renal cell carcinoma (RCC) patients. Curative surgery for VTT is high-risk, but neoadjuvant therapy may improve outcomes. The NAXIVA trial demonstrated a 35% VTT response rate after 8 weeks of neoadjuvant axitinib, a VEGFR-directed therapy. However, understanding non-response is critical for better treatment. Here we show that response to axitinib in this setting is characterised by a distinct and predictable set of features. We conduct a multiparametric investigation of samples collected during NAXIVA using digital pathology, flow cytometry, plasma cytokine profiling and RNA sequencing. Responders have higher baseline microvessel density and increased induction of VEGF-A and PlGF during treatment. A multi-modal machine learning model integrating features predict response with an AUC of 0.868, improving to 0.945 when using features from week 3. Key predictive features include plasma CCL17 and IL-12. These findings may guide future treatment strategies for VTT, improving the clinical management of this challenging scenario.

Venous tumour thrombus (VTT) occurs in 10%–15% of patients with clear cell renal cell carcinoma (ccRCC), where the primary tumour invades the renal vein and inferior vena cava (IVC) and can reach the liver and heart[1,2]. Whilst these patients are technically curable, the surgery required is extensive and complex, requiring multiple teams and the possibility of cardiopulmonary bypass[3,4]. There is considerable morbidity and mortality associated with surgery (5%–15%), which increases with the height of the VTT. If left untreated, RCC with VTT has a median survival of 5 months[1,3].

The NAXIVA trial (Phase II Neoadjuvant Study of Axitinib for Reducing Extent of Venous Tumour Thrombus in Renal Cancer with Venous Invasion, NCT03494816) was a phase II, single-arm,

[1]Early Cancer Institute, University of Cambridge, Cambridge, UK. [2]Department of Oncology, University of Cambridge, Cambridge, UK. [3]Cancer Research UK Cambridge Centre, University of Cambridge, Cambridge, UK. [4]Cambridge University Hospitals NHS Foundation Trust, Cambridge, UK. [5]Cancer Research UK Cambridge Institute, University of Cambridge, Cambridge, UK. [6]Department of Radiology, University of Cambridge, Cambridge, UK. [7]Department of Surgery, University of Cambridge, Cambridge, UK. [8]Translational Medical Sciences, School of Medicine, University of Nottingham Biodiscovery Institute, Nottingham, UK. [9]Warwick Medical School, University of Warwick, Coventry, UK. [10]Royal Devon University Healthcare NHS Foundation Trust, Exeter, UK. [32]These authors contributed equally: Rebecca Wray, Hania Paverd. [33]These authors jointly supervised this work: Mireia Crispin-Ortuzar, Grant D. Stewart, James O. Jones. *A list of authors and their affiliations appears at the end of the paper. ✉e-mail: joj21@cam.ac.uk

multicentre study investigating the use of neoadjuvant axitinib, a vascular endothelial growth factor receptor (VEGFR)-directed tyrosine kinase inhibitor (TKI), to reduce the ccRCC VTT. Loss of the tumour suppressor Von Hippel–Lindau (*VHL*) in ccRCC tumours activates the hypoxia response pathway of the cell, leading to induction of VEGF-mediated angiogenesis[5]. VEGFR-TKIs, either alone or in combination with immunotherapy, are used as first-line therapy for advanced RCC and have proven efficacy in patients with metastatic disease[6]. In the NAXIVA trial, 35% of patients experienced a reduction in VTT length of >30% after axitinib treatment, leading to less invasive surgery[7,8]. The remaining patients did not benefit from the neoadjuvant treatment. Progress is needed in understanding the reasons for non-response to improve treatments for these patients.

Little is known about the mechanisms driving treatment response in VTT. There is evidence that the VTT arises as a rapid outgrowth of the primary tumour and has shared driver events[9]. Studies have shown viable proliferating tumour cells[10] and immune infiltrate[11,12] in the VTT. Another recent study has shown that response to combination therapy in a VTT setting is increased when there are reduced T-helper cells in the pre-treatment tumour (NEOTAX, ChiCTR2000030405)[13]. At least one study is currently investigating combination treatment in the specific setting of ccRCC with VTT (NEOPAX, NCT05969496)[14]. However, no other prospective studies have examined the determinants of VTT response to systemic treatment.

In metastatic ccRCC, RNA-based signatures have been used to group patients according to the therapies most likely to benefit them[15–17]. A prospective study (BIONIKK, NCT02960906) classified patients into four transcriptome-based groups, finding that patients with an immune-low tumour microenvironment (TME) had improved survival during the combination of two immunotherapy drugs compared to patients with higher immune infiltration and inflammatory markers[18]. DNA, protein and clinical markers have also been investigated for patient stratification[19], but none have been widely adopted for use in metastatic cases, nor in neoadjuvant settings[20–22]. In wider oncology practice, the complexity of the TME and the disparate data generated from multiple sources complicate the development of predictive signatures. In other tumour types, the use of machine learning (ML) approaches to integrate data streams has provided valuable insights[23,24].

To investigate predictive markers of VTT response to axitinib, we performed a comprehensive multiparametric assessment of the TME and peripheral blood for patients in the NAXIVA trial. An ML model was then used to identify biomarkers for VTT response. Identifying reliable predictors of VTT treatment response would allow a personalised approach to treatment selection, which would improve outcomes, avoid overtreatment, and inform the design of future studies for VTT management.

## Results

### Assessment of VTT length response on the NAXIVA trial

The design and clinical outcomes of the NAXIVA trial have been fully reported elsewhere[7]. Briefly, eligible patients underwent baseline tumour biopsy, then up to 8 weeks of axitinib treatment, followed by surgery to remove the primary tumour and VTT (Fig. 1a). Serial blood samples were collected during the study. Extent of the VTT was assessed by an MRI scan at baseline (week 1), week 3 and week 9. For the present study, we included the 20 evaluable patients who were assessed in the trial.

In the main trial analysis, the primary endpoint was the change in the Mayo level of VTT[7]. However, the Mayo level is a categorical classification based on anatomical landmarks[8], and relatively small changes in the VTT dimensions may result in a change in Mayo level, or conversely, a large change may not cross a Mayo level. To investigate the TME biological response to therapy, we analysed against the continuous percentage change in the VTT length. 7/20 patients achieved a

>30% reduction in VTT length by week 9, and we classified this group as responders for our analysis (Figs. 1b, S1a, b). Considering the main clinical parameters, in keeping with the results reported in the clinical study[7], axitinib dosing, sex and TNM status did not appear to affect the VTT length change after treatment (Fig. S1c–f).

### TME of untreated VTT resembles the primary RCC TME

First, we assessed the microenvironment of untreated resected VTTs, outside of the NAXIVA trial, in comparison to the corresponding primary tumour. In keeping with previous studies, VTT consisted mainly of Carbonic Anhydrase 9 (CA9) positive viable tumour cells which filled the vessel lumen (Fig. 1c). In some examples, the morphology of the VTT was very similar to the primary (such as the cystic structures seen in Figs. 1c, S2a, b). There were extensive microvascular structures within the VTT, with CD31+ vessels surrounded by alpha smooth muscle actin (SMA) positive pericytes, between the CA9+ tumour cells (Figs. 1d, S2a, c). There was immune infiltration of both CD3+ T-cells and CD68+ macrophages (Figs. 1e, S2d). We assessed the relationship between the VTT TME and corresponding primary tumour by quantitative immunohistochemistry (IHC) for the following key markers in 10 paired cases: Ki67, CD8 and CD31 (Fig. S1g–i). The levels of Ki67, CD8 and CD31 were all significantly correlated between the VTT and primary tumour (CD8 $p = 0.022$, Ki67 $p = 4 \times 10^{-4}$, CD31 $p = 0.032$, Fig. S1g–i). These data demonstrate that the microenvironment of untreated VTT closely resembles that of its parent tumour, and so therapies that are effective against a primary should be effective against VTT.

### Higher microvessel density is associated with VTT response to axitinib

To analyse the effect of axitinib on tumour vasculature, whole slide imaging (WSI) of NAXIVA patients' baseline biopsy, and post-treatment VTT and primary tumour samples were analysed for microvessel density (MVD) by HALO image analysis (Fig. 2a). The baseline biopsy CD31+/CD34+ MVD in responders was significantly higher than in non-responders ($p = 7.88 \times 10^{-4}$, Fig. 2b), and was followed by a significant MVD reduction in the VTT after treatment ($p = 4.06 \times 10^{-4}$). In contrast, the MVD remained at a stable, low level in non-responders (Fig. 2b). This effect was also seen when quantifying CD31+ and CD34+ monomarkers (Fig. S3a, b). Upon assessment of SMA+ cancer-associated fibroblast (CAF) area coverage, there was a non-significant trend towards a reduction in the SMA+ CAF in non-responders on treatment ($p = 0.0867$, Fig. S3c).

### Circulating angiogenic factors are differentially induced in responders and non-responders

Axitinib inhibits the signalling response of the VEGF receptors to their soluble ligands. Therefore, we assessed the plasma levels of circulating angiogenesis markers during the trial. Absolute levels of VEGF-A were not different either before or during treatment in responders and non-responders (Fig. S4a). However, the fold change relative to each individual patient baseline showed circulating VEGF-A levels increased significantly by the end of treatment in responders compared to non-responders ($p = 0.0118$ at week 7, Fig. 2c). Absolute placental growth factor (PlGF) levels were low at baseline in both groups (Fig. S4b), followed by a strong induction in the responders at week 3 and a return to low levels after treatment ended and the tumour was resected. In fact, there was an approximately 7-fold PlGF increase in responders by week 3 of treatment ($p = 3.38 \times 10^{-3}$, Fig. 2d). There were some differences in early levels of additional angiogenic markers (Fig. S4), with VEGF-C higher in non-responders at baseline ($p = 0.0356$) and soluble VEGFR1 (sVEGFR1) higher in responders (at week 3, $p = 0.0445$). These markers seemed relatively stable on treatment (Fig. S4).

We then assessed the sources of the key identified angiogenesis markers in a published single-cell RNA sequencing dataset of

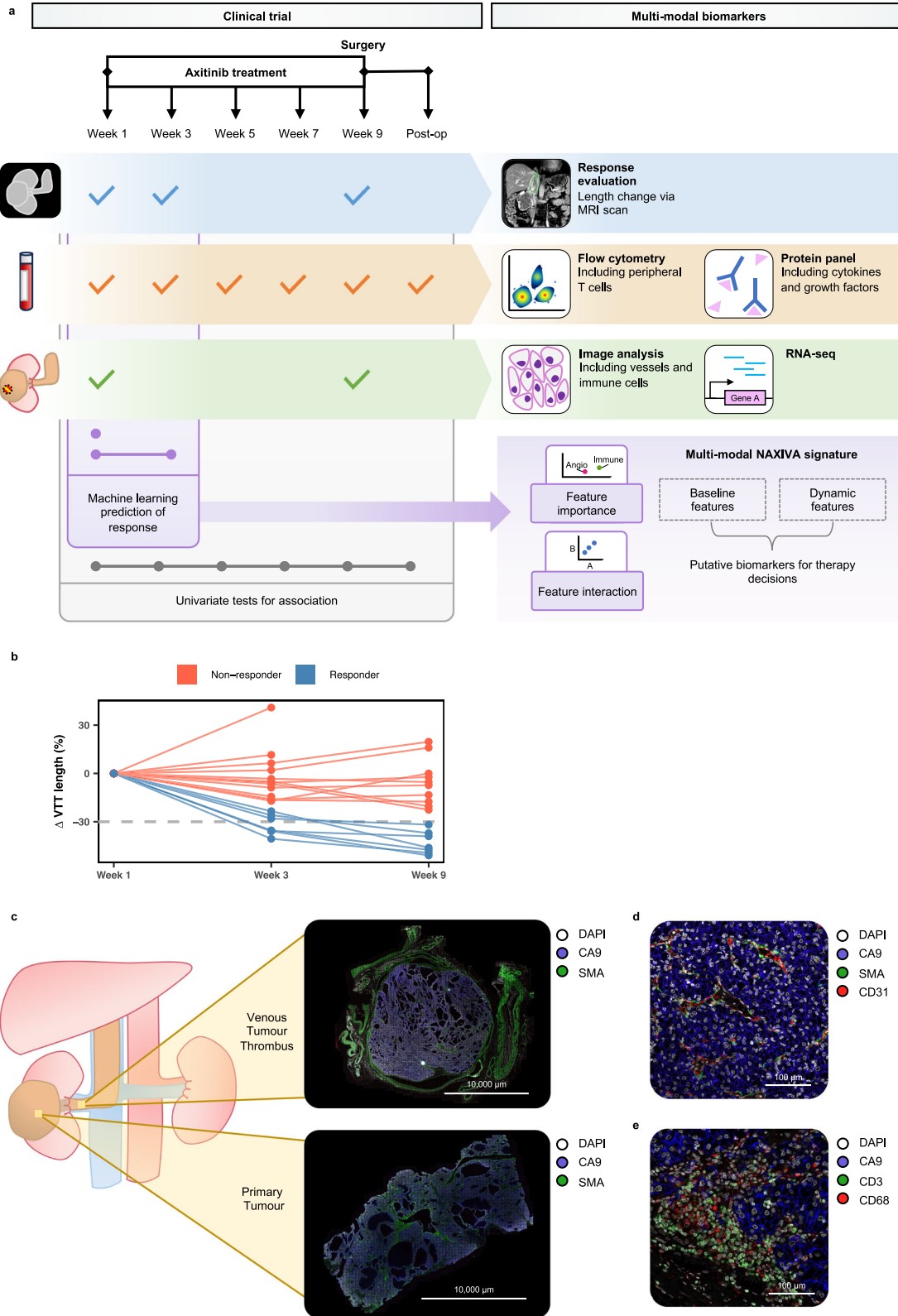

**Fig. 1 | Multiparametric investigation of VTT response in the NAXIVA trial.**
**a** Patients received up to 8 weeks of axitinib treatment. VTT response was evaluated by MRI at baseline, week 3 and week 9. Tissue was collected at baseline biopsy and at surgery from the VTT and primary tumour. Serial blood samples were taken before, during and after treatment. Research samples were assessed by a range of techniques to identify markers of response. Baseline and week 3 parameters were combined in a machine learning model for treatment response. **b** Patients reaching 30% reduction in VTT length by the end of the treatment course were classed as responders in the NAXIVA trial. 7 of 20 patients were classed as responders. **c**–**e** Whole slide scans of VTT and paired primary tumour; representative images from five paired cases are shown. **c** CA9+ viable tumour filled the lumen of the renal vein. **d** CD31+ microvessels surrounded by SMA+ pericytes were abundant within the VTT TME. **e** CD3+ T cells and CD68+ macrophages were present within the VTT TME.

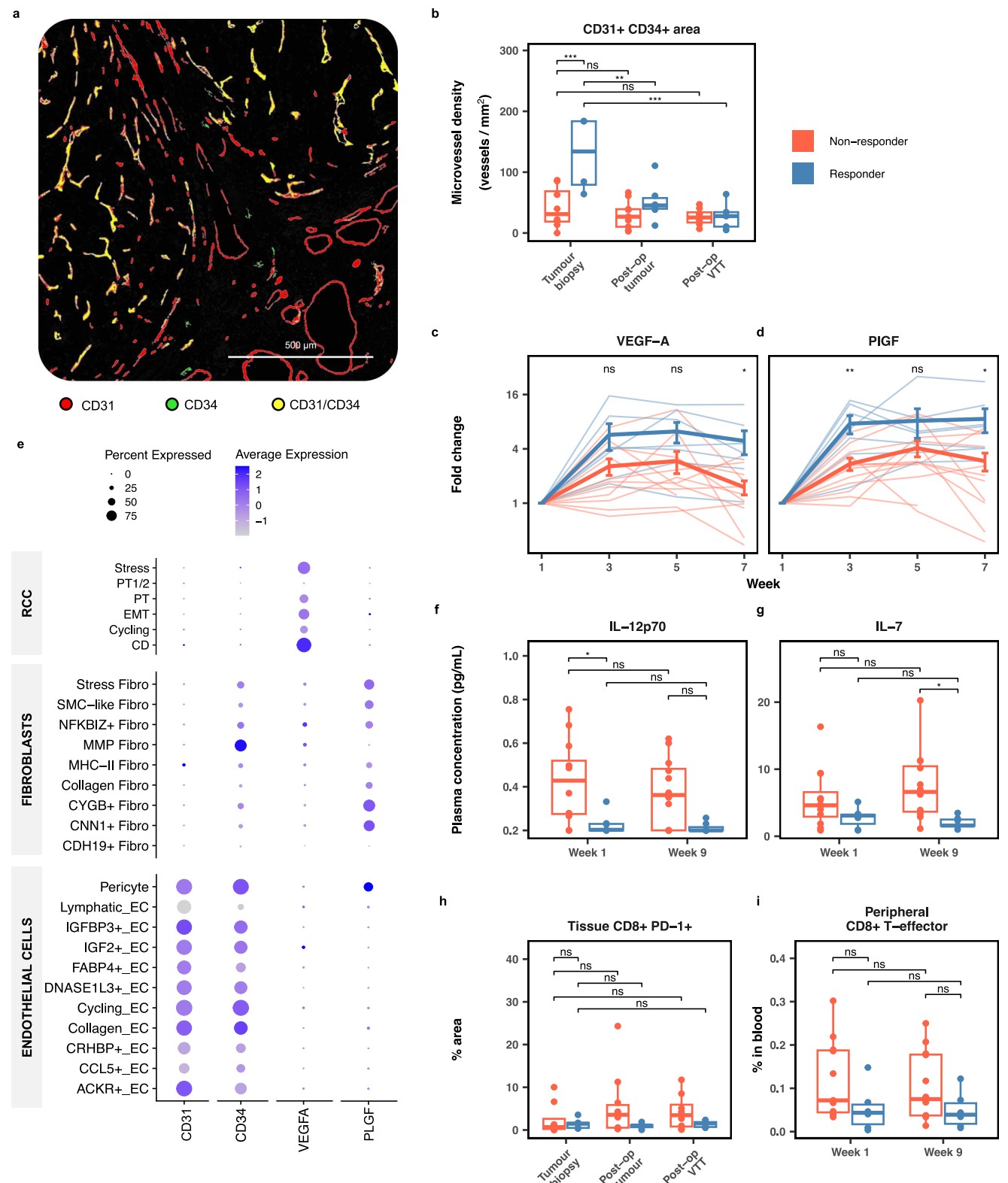

untreated RCC cases[25], which revealed the primary source of VEGF-A to be the cancer cells. In contrast, PlGF is made by SMA+ myofibroblast subsets and pericytes in the TME (Fig. 2e).

### Non-responders have an immune shift toward CD8+ T-cell immunity

Immune features may predict treatment response in advanced RCC[16–18]. We assessed the influence of tissue and blood immune

components on VTT response, including the plasma levels of immune cytokines. Circulating IL-12p70 levels were significantly higher in non-responders at baseline ($p = 0.0282$, Fig. 2f). IL-7 levels were significantly higher in non-responders after treatment ($p = 0.0344$, Fig. 2g). There was no difference in interferon gamma or any other cytokines assessed pre- and post-treatment (Fig. S5).

WSI of biopsy, VTT and primary tumour was analysed by HALO image analysis for T-cell subsets. Comparing responders with

**Fig. 2 | Responder and non-responder phenotypes. a** Representative image of HALO analysis markup of microvessels on multiplex immunofluorescence slides. **b** Responders had higher CD31+/CD34+ microvessel density pre-treatment than non-responders (one-way ANOVA with Tukey's post-hoc test; $p = 7.88 \times 10^{-4}$ for responder vs non-responder tumour biopsies, $p = 6.76 \times 10^{-3}$ for responder post-op tumour vs tumour biopsy and $p = 4.06 \times 10^{-4}$ for responder post-op VTT vs tumour biopsy; $n = 12$ tumour biopsies [4 responders, 8 non-responders], 15 post-op tumour samples [6 responders, 9 non-responders] and 13 post-op VTT samples [5 responders, 8 non-responders]). **c, d** Fold change in plasma VEGF-A (**c**) and PlGF (**d**) relative to pre-treatment baseline (thin lines, individuals; bold lines, mean and standard error of the mean; unpaired two-sided Student's $t$-test for responder to non-responder comparisons; $p = 0.0118$ for VEGF-A week 7, $p = 3.38 \times 10^{-3}$ for PlGF week 3 and $p = 0.0203$ for PlGF week 7; $n = 19$ for weeks 1–5 [7 responders, 12 non-responders], $n = 18$ for week 7 [7 responders, 11 non-responders]). **e** Single-cell RNA sequencing analysis of 12 untreated clear cell RCC showing expression of key angiogenesis genes by cell subset. **f, g** Responders had lower levels of IL-12p70 and IL-7 pre-treatment than non-responders (one-way ANOVA with Tukey's post-hoc test; $p = 0.0282$ for IL-12p70 week 1 comparison and $p = 0.0344$ for IL-7 week 9 comparison; $n = 19$ [7 responders, 12 non-responders]). **h, i** Non-responders trended towards higher immune markers in the blood ($n = 17$ week 1 samples [6 responders, 11 non-responders] and 18 week 9 samples [6 responders, 12 non-responders]) and tissue ($n = 13$ tumour biopsies [5 responders, 8 non-responders], 16 post-op tumour samples [6 responders, 10 non-responders] and 13 post-op VTT samples [5 responders, 8 non-responders]) (one-way ANOVA with Tukey's post-hoc test). All boxplots show the median (centre line), upper and lower quartiles (box bounds) and whiskers extending to 1.5× interquartile range. The source data for this figure are provided in the Source Data file. ns: $p > 0.05$, *$p \leq 0.05$, **$p \leq 0.01$, ***$p \leq 0.001$.

non-responders, no significant differences were observed in baseline biopsy CD8+ T-cell levels or CD8+ subsets, including in the CD8+/PD-1+ compartment (Fig. 2h), and this remained stable during treatment in both groups (Fig. S6). No significant differences were seen in overall CD4+ T-cells, CD4+/Foxp3+ T-regs, or CD68+ macrophages before or after treatment (Fig. S6).

Peripheral blood T-cell subsets were assessed by flow cytometry (Fig. S7a). There was a trend towards increased CD8+ T-cell levels in the peripheral blood of non-responders at baseline ($p = 0.294$, Fig. 2i), and a corresponding shift in the CD4+ to CD8+ T-cell ratio (Fig. S7a). Levels of other CD8+ and CD4+ subsets were similar between groups (Fig. S7a), as were levels of natural killer cells (Fig. S7b) and monocyte subsets (Fig. S7c). There were no differences in B-cell subsets or plasmacytoid dendritic cells (Fig. S8).

## Responders and non-responders have distinct transcriptomic profiles

RNA-seq was performed on baseline biopsies to investigate transcriptomic predictors of response, with 13 biopsies meeting our quality criteria (see "Methods" section). Principal component analysis of baseline biopsies demonstrated clustering of responders and non-responders (Fig. 3a). Differential gene expression analysis identified some immune-related transcriptomic differences, such as *IL12RB2* (IL-12 receptor beta subunit) and *ARG2* (Arginase type II) (Fig. 3b). However, Gene Ontology (GO) analysis showed that the majority of the most differentially expressed genes are in metabolic pathways, with examples including *ALDOB* (Aldolase B) and *ACSBG1* (Acyl-CoA Synthetase, Bubblegum Family, member 1) (Fig. S9a). Seven were solute carrier (SLC) family genes, four of which reached high significance ($p < 0.001$). ccRCC survival data from The Cancer Genome Atlas[26] suggests high expression of *SLC6A19* (Sodium-dependent neutral amino acid transporter B(0)AT1), *SLC22A12* (Solute carrier family 22 [organic anion/cation transporter], member 12) and *SLCO2A1* (Solute carrier organic anion transporter family member 2A1 - a prostaglandin transporter) are associated with improved overall survival, as is *ALDOB* (Fig. S9b).

The most differentially expressed genes from NAXIVA were mapped onto publicly available data from a Phase III study, IMmotion151, which described seven distinct molecular clusters derived from pre-treatment tumour transcriptomes in advanced RCC[17]. Genes highly expressed in NAXIVA responders were also highly expressed in IMmotion151 clusters C1 (angiogenesis/stromal) and C2 (angiogenesis), including the *SLC* family members. In contrast, patients in clusters C4 (T-effector/proliferative), C5 (proliferative) and C6 (stromal/proliferative) have lower expression of the genes highly expressed in NAXIVA responders, and higher expression of the genes highly expressed in NAXIVA non-responders (Fig. 3c).

We generated signatures from the top differentially expressed genes in the NAXIVA patients. We applied these signatures to published patient-level RNA-seq data from the Javelin Renal 101 study[16], a randomised phase III clinical trial comparing axitinib plus avelumab with sunitinib in advanced renal cancer. We stratified the patients into NAX-RNA-hi and NAX-RNA-lo groups. Progression-free survival (PFS) was compared between patients in the NAX-RNA-hi and NAX-RNA-lo groups in each arm of the Javelin trial (Fig. 3d, e). For the sunitinib arm, the patients in the NAX-RNA-hi group had a higher PFS than those in the NAX-RNA-lo group (Fig. 3d). However, this effect was not true in the avelumab + axitinib arm of the trial (Fig. 3e). Published RNA-based predictive signatures of RCC treatment response to anti-angiogenic or immunotherapy from IMmotion151 and Javelin Renal 101 were used to calculate scores for each patient in NAXIVA (Fig. 3f, g). NAXIVA responders scored higher in the Javelin Renal 101 Angio RNA signature[16], on average. Three of the non-responders scored highly in the Javelin Immuno score, but the spread of the scores is broad (Fig. 3f), which we expect, since the Javelin Immuno score is derived from patients treated with immunotherapy. The IMmotion151 molecular subset clusters used to differentiate patients in the ongoing Phase II OPTIC-RCC study[17,27] – C1/2 angio/stromal, and C4/5 T-effector/proliferative – again showed correlation with the NAXIVA responder and non-responder groups, with non-responders achieving a higher C4/5 score on average than responders (Fig. 3g). However, the C1/2 score was less able to differentiate between these patients.

## A machine learning model integrating multiple baseline features predicts treatment response

Integrating multiple data strands into a predictive model may provide better insights into the drivers of response in oncology trials[23,24]. We developed an ML approach using baseline (pre-treatment) features to predict response outcome, considering a binary classification of response as above, where response is defined as a >30% reduction in VTT length compared to baseline. The input data consisted of 62 features measured for each of the 20 patients. To reduce overfitting of the model to the small dataset, highly correlated features were reduced as part of data pre-processing, and the first part of the model involves a dimensionality reduction step, which selects the features contributing most to response (Fig. 4a). In this model (baseline model), best performance was achieved when three features of the 62 were selected (Table S1). A logistic regression was then fitted to the reduced dataset. We used a leave-one-out cross-validation approach, whereby the feature selection and model training were repeated for each group of 19 patients, generating 20 models which each predicted the response of the remaining one patient. While the number of patients in the study was relatively low, this approach prioritised feature identification for further investigation and follow-up in future studies.

The scaled data input to the model are shown in Fig. 4b. With these baseline features, the model achieved an area under the receiver operating characteristic curve (AUC) of 0.868 (Fig. 4c, Table 1). Three specific features were selected repeatedly in at least 8 of the 20 iterations of the model (Table 2). These were found to be plasma IL-12p70, CCL17, and microvessel density. CCL17 was not identified in the

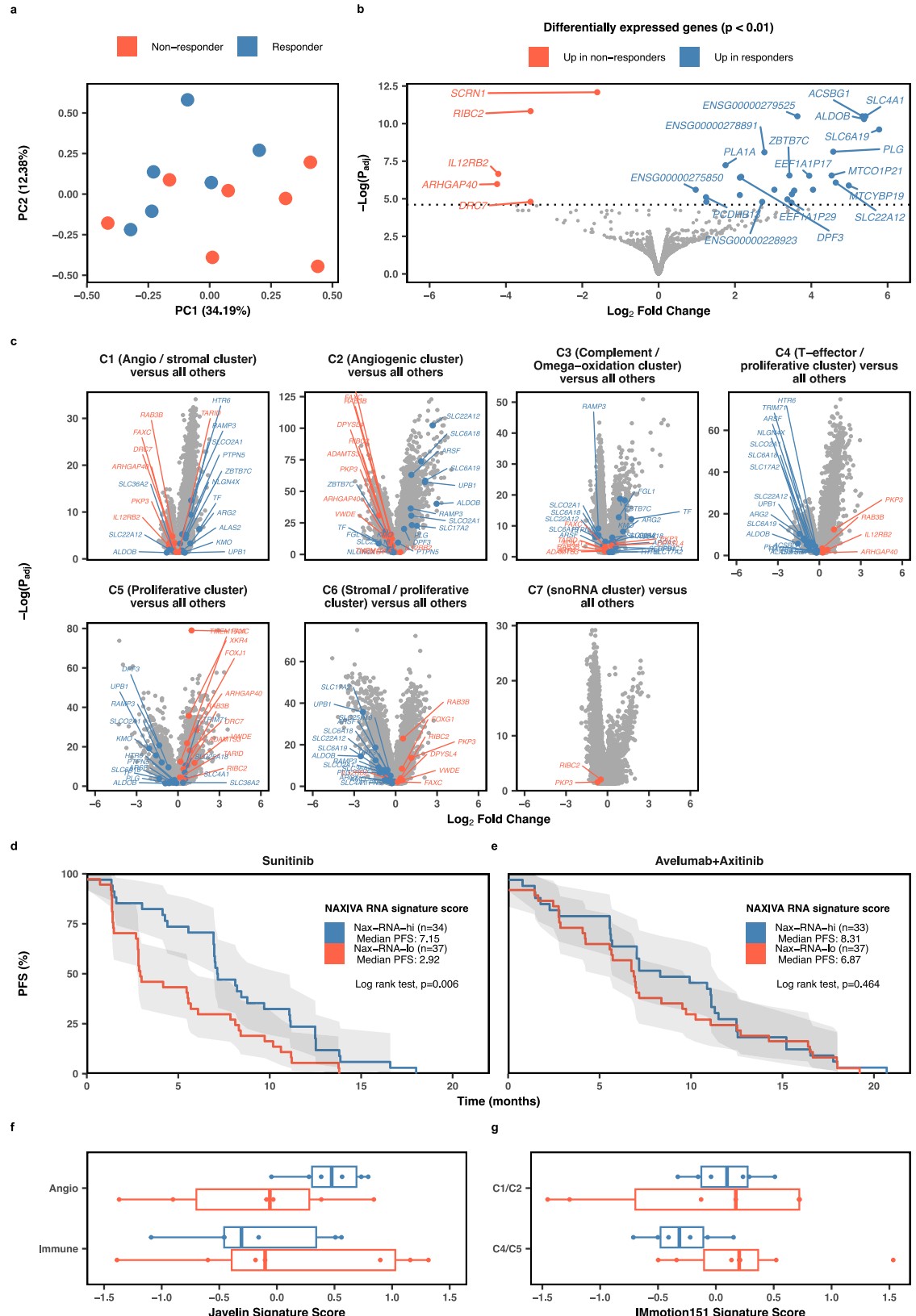

univariate data analysis (Fig. S5) but had the highest selection frequency and relative weight assigned by the logistic regression. Responders were always low in circulating CCL17 and IL-12p70, and generally, but not exclusively, higher in tumour MVD (Fig. 4d). Published scRNA-seq data show that CCL17 is not expressed by RCC cells but may be expressed by conventional dendritic cells in the TME (Fig. S10). The CCL17 receptor,

CCR4, is expressed on CD4+ T-cells and is highest on CD4+ T-reg cells. IL-12 receptors are widely expressed on T-cell and NK populations (Fig. S10). The expression of the genes encoding these baseline features (*PECAM1, CD34, CCL17, CCR4, IL12A, IL12B*) identified by the model were able to stratify the patients in Javelin Renal 101[16] according to progression-free survival (log-rank test, $p = 0.034$, Fig. S11a).

**Fig. 3 | RNA-seq analysis of baseline biopsies. a** PCA plot of RNA-seq data for pre-treatment biopsies of responder ($n = 6$) and non-responder ($n = 7$) tumours. **b** RNA-seq results comparing responder to non-responder biopsies via DESeq2. Labelled points are $p < 0.01$. Differential expression analysis was performed using DESeq2, applying a two-sided Wald test. *P*-values were adjusted for multiple comparisons using the Benjamini–Hochberg method to control the false discovery rate (FDR). Data are shown as $Log_2$ fold changes with associated adjusted *p*-values. **c** Most differentially expressed genes ($p < 0.05$) plotted on the IMmotion151 RNA-seq clusters. Data, including statistical analysis, were directly extracted from the original study[17]. **d**, **e** The most differentially expressed genes in NAXIVA ($p < 0.05$) stratified patients according to PFS in the Javelin Renal 101 study[16] for the sunitinib arm (**d**) and not for the avelumab + axitinib arm (**e**). Grey shaded areas indicate the 95% confidence interval. **f** RNA signature scores for the NAXIVA patients in the transcriptomic signature identified in the Javelin Renal 101 study[16] ($n = 6$ responders, 7 non-responders). **g** RNA signature scores for the NAXIVA patients in the transcriptomic signature identified in the IMmotion151 study[17] ($n = 6$ responders, 7 non-responders). All boxplots show the median (centre line), upper and lower quartiles (box bounds) and whiskers extending to 1.5× interquartile range.

## Adding early dynamic measures improves the performance of the machine learning model

A challenge in biomarker development for all cancers is the inherent variability between patients, either due to tumour differences or their underlying physiology. Early markers of response after treatment has begun may be informative for clinical decision making. Therefore, we updated the model to include measurements of fold changes in the plasma angiogenic factors after three weeks of axitinib treatment (Fig. S11b, c). This second model (dynamic model) achieved a higher AUC of 0.945 (Fig. 4c), with high selection of CCL17 and IL-12p70, as before (Table 2, Fig. 4e). Interestingly, the week 3 fold change in plasma sTie-2 and PlGF also showed potential for stratification. The baseline and dynamic models were capable of predicting surgically relevant Mayo Classification-based response in these patients (Fig. S11d).

Comparing the longitudinal performance of the models, the dynamic model returned higher confidence in the responder classification, which we interpret as a higher probability of response, giving a score of >0.5 for all seven responders (and five of them >0.9). The baseline model gave six responders >0.5 (and three of them >0.9), with one misclassified as <0.1 (Fig. 4f). The selected features corresponded as expected to the molecular clusters in the IMmotion151 RNA-seq data (Fig. 4g). PlGF was much higher in the angio-stromal cluster, C1, in keeping with its stromal origin in the single-cell analysis.

## Discussion

Our study draws on a unique sample set from a phase II clinical trial, with tissue and serial blood samples taken before, during and after treatment. We conducted a multiparametric analysis including tissue factors by digital pathology and RNA-seq, and both cell-based and soluble-factor analysis of peripheral blood. Furthermore, we deployed ML approaches to prioritise and integrate the parameters. This allowed us to gain new insights into determinants of response in ccRCC with VTT, a challenging clinical scenario.

The highly organised nature of the VTT TME was striking, with an established microvessel network, stroma, and immune infiltrate as seen in primary ccRCC. High baseline MVD was predictive of response to neoadjuvant axitinib, particularly in the CD31+/CD34+ subset of vessels. A good response was also associated with greater induction of angiogenic growth factors, particularly stroma-derived PlGF. Non-responders had an immune-high phenotype, with higher levels of IL-7 and IL-12, and trends to increased circulating CD8+ T-effectors in blood and CD8+/PD-1+ T-cells in the TME. Assessing transcriptomic data from baseline biopsies, several genes were associated with good response, notably in the solute carrier gene family. The ML model selected IL-12p70, CCL17 and biopsy MVD for response prediction, with model performance improvements seen after the inclusion of early response data, selecting for sTie-2 and PlGF induction. These identified features could be readily assayed in clinical practice.

Our data support previous reports that the VTT is closely related to the parent tumour and is essentially a primary tumour existing within the lumen of the vessel[10–12]. Our finding that VTT axitinib responders have a pro-angiogenic, immune-low phenotype is in keeping with observations in the metastatic setting, where an angiogenesis-rich subgroup is proposed to benefit from VEGFR-TKI therapy[15–18]. Amongst circulating factors, PlGF has previously been described as a pharmacodynamic marker for TKI treatment[28,29]; however, it has not previously been found to be a predictive marker for therapy outcome as demonstrated here. IL-7 supports lymphocyte proliferation[30], IL-12 is critical for cytotoxic T-cell differentiation[31], and there is evidence these cytokines may cooperate to enhance anti-tumour immunity[32].

Our study finds that the expression of highly upregulated proteins in NAXIVA responders was also upregulated in C1/2 patients (angiogenic/angio-stromal) in publicly available IMmotion151 transcriptomic data, whereas the opposite was observed for C4/5/6 (T-effector/proliferative, proliferative and stromal/proliferative), which fitted better with the non-responding patients. We also find that the genes encoding our blood markers were able to stratify the patients in the Javelin Renal 101 trial according to progression-free survival. When published RNA signatures were applied to the NAXIVA transcriptomic data, we found some correlation with the outcome, particularly for the Javelin Renal 101 Angio score.

The RNA-seq data from the patient biopsies revealed an association between response to axitinib and a number of genes involved in several metabolic pathways, including genes in the solute carrier family, namely *SLC6A19, SLC22A12, SLCO2A1* and *SLC4A1*. The concomitant increase in MVD and expression of genes related to solute metabolism and transport point towards a relationship between the metabolic pathways in the tumour and the induction of angiogenesis. For example, *SLCO2A1* (a prostaglandin transporter) may regulate the endothelial response to prostaglandins[33], influencing angiogenesis and potentially responsiveness to anti-angiogenic therapy. Three of the SLC family members identified in NAXIVA responders were associated with a favourable prognosis in TCGA data, where they were also predictive of immune microenvironment and drug response[34]. Considering other upregulated metabolic genes, *ALDOB* has also been reported to have prognostic significance in RCC[35]. These genes are expressed by normal renal tubules, so they may mark well-differentiated, less aggressive tumours.

A further question in the treatment of metastatic ccRCC is the potential synergistic effect of combining immunotherapy with VEGFR-directed TKIs, where the TKI is proposed to boost the effect of immunotherapy. Pre-clinical data indicate that VEGFR-TKIs enhance immunity by a variety of effects, including the reduction of immune suppressive myeloid cells in the tumour microenvironment (TME)[36–39]. The recent NEOTAX study found that responders to neoadjuvant toripalimab plus axitinib for ccRCC patients with VTT had lower densities of CD4+ T-helper cells in the tumour biopsy[13]. CD4+ T cells are the predominant cell type expressing the CCL17 receptor, CCR4 (Fig. S10a). However, in our data, we did not find any clear evidence of axitinib altering the immune profile and the overall immune phenotype remained stable on treatment. This is consistent with a study of neoadjuvant pazopanib in localised RCC, which did not find any change in immune signatures on treatment[21]. Axitinib has a narrow range of targets compared to other TKIs used in RCC[40], so these observations do not rule out an immune modulatory effect of TKIs that target a wider range of receptors, such as lenvatinib or cabozantinib.

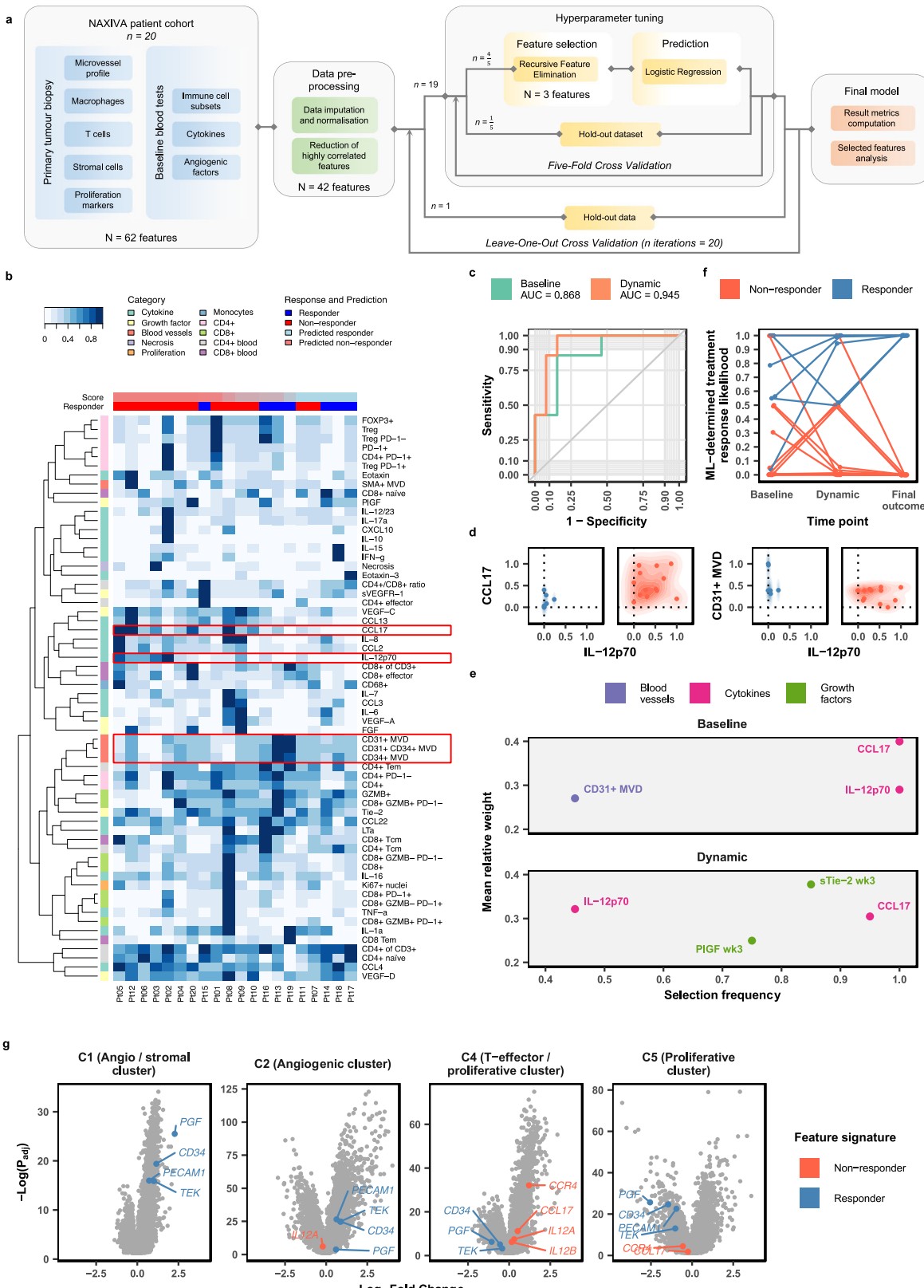

**Fig. 4 | Machine learning model predicts response to axitinib. a** Machine learning model workflow. **b** Pre-processed data description and model-predicted scores for each patient. **c** Receiver operating characteristic curve. **d** Selection frequency for selected features (the number of times the feature was selected across the leave-one-out cross-validation iterations divided by the total number of iterations). **e** Density plots of scaled values of two features with the highest selection frequency for responders and non-responders. **f** Prediction of response increased in accuracy and confidence when week 3 measurements are included in the analysis. **g** Signature from NAXIVA blood data displayed on IMmotion151 RNA-seq data. Data, including statistical analysis, were directly extracted from the original study[17]. The source data for this figure are provided in the Source Data file.

**Table 1 | Results for the ML models**

| Metric | Baseline model | | Dynamic model |
|---|---|---|---|
| AUC ROC | 0.868 | < | 0.945 |
| Accuracy | 0.850 | = | 0.850 |
| Sensitivity | 0.857 | = | 0.857 |
| Precision | 0.750 | = | 0.750 |
| F1-score | 0.800 | = | 0.800 |
| Specificity | 0.846 | < | 0.923 |
| AU PRC | 0.796 | < | 0.886 |

**Table 2 | Most significant features identified in the ML models**

| Marker | Sample | Timepoint | Selected in base-line model | Selected in dynamic model |
|---|---|---|---|---|
| CD31 | Tissue | Week 1 | * | |
| CD34 | Tissue | Week 1 | * | |
| IL-12p70 | Blood | Week 1 | * | * |
| CCL17 | Blood | Week 1 | * | * |
| PlGF | Blood | Week 3 | | * |
| sTie-2 | Blood | Week 3 | | * |

Induction of PlGF at week 3 was a key marker of a good outcome in NAXIVA. Biological heterogeneity, both between patients and within tumours, is a challenge for the development of baseline predictive biomarkers, whose limited performance could be surpassed by the dynamic measurement of blood markers such as PlGF. Early blood biomarker changes may have significant clinical utility as they are readily assayed in the clinic. In the scenario of VTT management, it may give confidence in continuing with neoadjuvant therapy against proceeding directly to surgery. Our data do not provide a mechanism for the PlGF induction; however, we hypothesise that in responders, the effective blockade of the VEGFR axis induces a hypoxic response with increased production of VEGF-A and PlGF as a compensatory mechanism. It is not clear whether the responders and non-responders are biologically distinct in this respect or whether there is a spectrum of effects depending on the degree of VEGFR inhibition achieved. PlGF is reported to potentiate the effectiveness of VEGF signalling, and so it may be a mechanism to overcome blockade[41], particularly important in pathological angiogenesis compared to physiological angiogenesis[42]. PlGF is an attractive marker for further exploration as there is an existing clinical assay used for pre-eclampsia, which might be re-purposed.

A challenge in the analysis of small clinical trial datasets with extensive translational analysis is the large number of parameters assessed for predictive value in comparison to the number of patients enrolled in the study. ML approaches may enhance the analysis of similar datasets. The ML model based on baseline features achieved good performance for response prediction on an internal validation set. Performance is enhanced by data from week 3, again demonstrating the potential value of dynamic marker assessment. The models were limited by a small dataset, but our approach (leave-one-out nested cross-validation with consensus-based feature importance) effectively maximised the training data and tested the model on independent samples. The discovery of a robust predictor with a limited and highly stable set of features indicated a strong signal and substantially reduced the risk of overfitting.

The models selected a small number of factors based on plasma and tissue measurements, which could be readily translated into the clinic. The result for CCL17 illustrates the utility of the ML approach, as this cytokine was assigned high priority by the ML models despite not being seen in our initial single parameter screens of the data. CCL17 is

an important regulator of T-cell immunity acting on CCR4 and has been shown to be negatively prognostic in RCC[43]. The ML approach provided some insight into the interaction between the different features, particularly with responders being low in both IL-12p70 and CCL17. Changes in sTie-2 were also important in the dynamic model, which could be an alternative pathway for angiogenesis[44]; however, we interpret this finding with caution: although the fold change was consistent, the absolute changes in sTie-2 concentration in each patient were small. RNA-seq data were not included in our ML model due to the potential for noise amplification of adding several thousand differentially expressed genes to the other parameters. We observed differences in RNA and protein results; for instance, CCL17 was undetected in our RNA-seq and present at low levels in published single-cell datasets. This may suggest low transcript expression in tumours, making detection challenging, or indicate the importance of a non-tumour source, such as primary or secondary lymphoid tissue.

The study is limited by the small size of the trial, with only 20 participants, and by the lack of an external validation set for the key parameters identified due to the unique nature of the study. We are restricted in both respects by the lower prevalence of VTT relative to all RCC cases; specific VTT management has been the subject of phase II trials to date, but a dedicated phase III VTT trial is likely unfeasible. Thus, we are limited to suggesting these markers as priorities for further work. Axitinib has been superseded by more active treatment combinations of TKIs and immunotherapy in the metastatic setting[45–47], which is now being explored in pre-operative trials[14]; nonetheless, the TKI monotherapy in NAXIVA provides a useful comparator to any translational data arising from these IO-TKI studies.

Beyond phase II trials of current treatments, newer agents such as more potent TKIs or bispecific immunotherapies may have application in improving oncologic outcomes for VTT patients. This must be balanced against the risks of toxicity. Our investigations of the microenvironment and blood features have identified predictive biomarkers that might be clinically and functionally validated in these studies, either alone or as a combined assay. It will be interesting to see whether the key features identified by our study, which mainly divide the patients into immune and angiogenic, are still valuable when combination treatment is used, or whether others emerge. It is critical that a range of translational analysis approaches, including tumour, blood, RNA and protein-based approaches linked to advanced cancer imaging, are built into future study designs to gain a full understanding of the mechanisms of tumour response and resistance.

## Methods
### Participants
NAXIVA was a single-arm, single-agent, phase II, open-label, multi-centre UK-based study (NCT03494816, UK ethical approval REC reference: 17/EE/0240). Full study details, including the trial protocol, have been previously published[7]. Key inclusion criteria included: age >18, T3a, T3b or T3c, N0/N1, M0/1, biopsy-proven clear cell RCC, suitable for immediate surgery. The baseline characteristics of the patients are summarised in the clinical publication[7]. Patients were treated with axitinib at a starting dose of 5 mg BD, escalated to 7 mg BD and then 10 mg BD every 2 weeks. The drug was stopped a minimum of 36 h and a maximum of 7 days before surgery. The 20 evaluable patients in the intention-to-treat population in the main trial are included in the current study. Additional samples from untreated RCC patients with VTT were obtained from the ARTIST study (NCT04060537, UK ethical approval REC reference: 20/EE/0200) and the DIAMOND study (UK ethical approval REC reference: 03/018). All patients were consented following GCP principles, and the nature and possible consequences of the studies were explained. The studies were performed in accordance with the Declaration of Helsinki.

## Response evaluation

The technique for measuring VTT length by MRI is detailed in the original clinical report[7], summarised as follows: Calculate the sum of (i) length of RV thrombus; (ii) the length of IVC tumour thrombus above the renal vein (measured from midpoint of the ostium of RV + IVC to tip of tumour thrombus); (iii) the length of IVC tumour thrombus below the renal vein (measured from midpoint of the ostium of RV + IVC to the tip of tumour thrombus). The percentage change in length at each timepoint (LT) compared to the length at baseline (LB) was calculated as (LT-LB)/LB*100.

## Histology & image analysis

Immunohistochemistry was performed on the Leica Bond III platform by standard automated procedure. The following antibodies were used: CD8 (4B11 Leica PA0183), CD31 (JC70A Leica PA0414), Ki67 (MIB-1 Dako M7240). For immunofluorescence, 3-micron formalin-fixed paraffin-embedded (FFPE) sections were dewaxed in xylene and rehydrated in graded alcohols. Heat-Induced Epitope Retrieval was performed in Tris-EDTA pH 9. After blocking, slides were incubated with primary antibodies at 4 °C overnight. Antibodies used were as follows: CD31 (JC/70A, Abcam ab9498), CD34 (R&D Systems AF7227), SMA (Abcam ab5694), CD68 (KP1, Invitrogen MA5-13324), Ki67 (EPR3610, Abcam ab92742), CD8 (SP16, Invitrogen MA5-14548), Granzyme B (Leica NCL-L-GRAN-B), PD-1 (R&D Systems AF1086), CD4 (EPR6855, Abcam ab133616), FOXP3 (236A/E7, Abcam ab20034), CA9 (R&D Systems AF2188), CD3 (D7A6E, Cell Signalling Technology 85061S). Samples were washed and incubated in fluorescently conjugated secondary antibodies. Nuclei were counterstained with DAPI. Whole slides were scanned at ×40 magnification on the Zeiss Axio Scan Z1 system. High-resolution images were acquired using a Leica SP5 Confocal Microscope at ×40 objective magnification.

Image analysis was performed using HALO Software (Indica Labs). The tumour area was outlined manually for all slides. Slides with inadequate tissue quality for quantification were excluded from the analysis. Pre-defined analysis settings were applied to all slides for objective quantification. Analysis algorithms as follows: HighPlex FL v3.1.0, Object Colocalization FL v1.0, Area Quantification FL v2.1.5, Area Quantification v2.4.3, Multiplex IHC v3.1.4.

## Flow cytometry

PBMC samples collected during the trial were thawed and re-suspended in X-VIVO complete media (Lonza). Fc receptor block was used (Miltenyi). Cells were stained using standardised antibody panels (Table S2). Viability was assessed by Zombie Aqua viability dye (Biolegend). Samples were run on a BD Symphony instrument. Appropriate single stain compensation bead controls were used. Data was analysed using FlowJo software.

## Cytokine arrays

Cytokine arrays were run by the Core Biochemical Assay Laboratory at the Cambridge Biomedical Research Centre, according to the manufacturer's instructions. The following kits were used from MesoScale Discovery: Human 10-plex Cytokine Panel 1 K15050D, Human 10-plex ProInflammatory Cytokine K15049D-2, Human 10-plex Chemokine Panel 1 K15047D, V-PLEX Angiogenesis Panel 1 Human Kit, K15050D. Plates were analysed on an MSD s600 instrument and results calculated by MSD Workbench software.

## Statistical analysis

Statistical analysis was conducted using R's ggpubr (v0.6.0) package. For two-way comparisons, the unpaired two-tailed Student $t$-test was used with Bonferroni multiplicity correction where appropriate. For multiple comparisons, one-way ANOVA was used with Tukey's post-hoc test. Pearson's correlation was used for correlation analysis. All boxplots: centre line, median; box limits, upper and lower quartiles; whiskers, largest / smallest value or 1.5× interquartile range.

## Single-cell analysis

To assess the expression of key genes in single cells derived from patients with RCC, we downloaded data from https://www.cell.com/cancer-cell/fulltext/S1535-6108(22)00548-7 via Mendeley Data: https://data.mendeley.com/datasets/g67bkbnhhg/1. To convert from .h5ad object to Seurat object, we used sceasy (https://github.com/cellgeni/sceasy), prior to normalisation by mitochondrial content with SCTransform from the R package Seurat (v4.3.0). The average expression and the percentage of cells that expressed the genes of interest were plotted. For clarity, we restricted cell types to endothelial, fibroblasts, and RCC cells as other cell types did not express the genes of interest (data not shown).

To plot the relative prevalence of cell types within different tissue compartments, we used code developed in https://www.cell.com/cancer-cell/fulltext/S1535-6108(22)00548-7 and documented in https://github.com/ruoyan-li/RCC-spatial-mapping. Briefly, we calculated the observed and expected number of cells of all cell types/subtypes across different tissues. Adrenal metastasis and tumour thrombus were excluded from this analysis as they were only sampled in single patients. We also excluded blood cells as no endothelial, fibroblasts, and RCC cells were expected in this compartment.

## RNA-seq

**RNA extraction & sequencing.** RNA was extracted from formalin-fixed paraffin-embedded (FFPE) tissue using the ReliaPrep FFPE Total RNA kit, according to the manufacturer's instructions, and assessed by Qubit and Agilent RNA ScreenTape System and for quantity and quality. We extracted sufficient RNA from 16 of the 20 samples. RNA library preparation was done using the Watchmaker Genomics RNA Library Prep Kit with Polaris Depletion, according to the manufacturer's instructions and running 18× PCR cycles for each sample. Indexing was done using the xGen™ Stubby Adaptor and UDI primers from Integrated DNA Technologies™, and sequencing via Illumina sequencing. The samples were run on an S4 flow cell on NovaSeq6000 with a read length of PE50. Manufacturer's instructions followed for the run, including spike-in of 1% PhiX.

**RNA-seq processing and analysis.** Reads were mapped using Salmon (v1.10.0) with GRCh38.p44 from Gencode. Samples were only included in the analysis if the sequencing duplication rate was <65%. Genes were included if the maximum count per biopsy was >10 and if more than 50% of the biopsies had counts >0. Counts were normalised by variance-stabilised transformation and the top 500 genes generated principal components for the PCA plot. DESeq2 was used to identify differential expression between responders and non-responders, and between post- and pre-treatment samples. Genes satisfying $P_{adj} < 0.05$ and absolute $Log_2$ Fold Change (LFC) > 2 were used in pathway analysis via GO in the Cluster Profiler R package (v4.12.0) and plotted using EnrichPlot (v1.24.0). The same genes, and the genes for the features identified in the ML models, were highlighted on the published RNA-seq differential expression analysis data from IMmotion151[17] to generate Figs. 3c and 4g, respectively.

**RNA signature scores.** The NAX-RNA scores were generated using the genes satisfying $P_{adj} < 0.05$ and absolute LFC > 2 in the NAXIVA transcriptome data. For each patient, a response signature was calculated as the mean expression of the genes where NAXIVA LFC > 2, and a non-response signature where LFC < −2. Javelin[16] patients with a response signature in the top quartile and a non-response signature in the bottom quartile were termed NAX-RNA-hi, and patients with response in the bottom quartile and non-response in the top quartile were termed

NAX-RNA-lo. Median PFS was compared between the two groups, and a log-rank test was run using Survival (v3.7.0) and Survminer (v0.5.0) R packages.

The genes used for the Javelin Renal 101 Angio signature[16]: *NRARP, RAMP2, ARHGEF15, VIP, NRXN3, KDR, SMAD6, KCNAB1, CALCRL, NOTCH4, AQP1, RAMP3, TEK, FLT1, GATA2, CACNB2, ECSCR, GJA5, ENPP2, CASQ2, PTPRB, TBX2, ATP1A2, CD34, HEY2, EDNRB*. The genes used for the Javelin Renal 101 Immuno signature: *CD3G, CD3E, CD8B, THEMIS, TRAT1, GRAP2, CD247, CD2, CD96, PRF1, CD6, IL7R, ITK, GPR18, EOMES, SIT1, NLRC3, CD244, KLRD1, SH2D1A, CCL5, XCL2, CST7, GFI1, KCNA3, PSTPIP1*. The genes used for IMmotion151 Angio (C1/C2) signature[17]: *VEGF-A, KDR, ESM1, PECAM1, ANGPTL4, CD34, FAP, FN1, COL5A1, COL5A2, POSTN, COL1A1, COL1A2, MMP2*. The genes used for IMmotion151 Immuno (C4/C5) signature: *CD8A, EOMES, PRF1, IFNG, CD274, CDK2, CDK4, CDK6, BUB1B, CCNE1, POLQ, AURKA, MKI67, CCNB2*. The genes used for the NAXIVA Angio signature: *PGF, TEK, PECAM1, CD34, VEGF-A*. The genes used for the NAXIVA Immuno signature: *CCL17, IL12A, IL12B, IL-7*. The counts were normalised by variance-stabilised transformation, and the mean and standard deviation were calculated for each gene. For each patient, the score per gene is (expression − mean expression) / standard deviation across all patients. The total signature score per patient is the mean of the scores for each gene.

## Machine learning models

**Training.** We created an ML framework to predict response to axitinib. We used leave-one-out cross-validation (LOOCV) on NAXIVA's 20 patients to train and optimise the models. We used an increasing number of features, starting with baseline biopsy and blood features (baseline model, 62 features, Fig. 4a), then adding growth factor fold-change features at week 3 of axitinib treatment (dynamic model, 69 features, Fig. S11b). For each combination, we retrained the framework and derived a new model. The full list of features can be found in Table S3. The models included recursive feature elimination (RFE) using a logistic regression estimator; the number of features selected by RFE was varied empirically, and several approaches of combining baseline and dynamic features were explored; the method achieving the highest AUC was selected ($N$ = 3 features, Table S1). The predictions were done by logistic regression with stochastic gradient descent, all coded in Python using scikit-learn (v1.4). Before entering the ML algorithm, all data underwent three pre-processing steps: iterative imputation, min-max standardisation and collinearity reduction. The estimator used for the iterative imputation was Bayesian Ridge. Collinearity reduction removed all features with a Spearman's Rank correlation above 0.75, retaining one feature at random from the collinear group. We used a five-fold cross-validation setup to optimise model hyperparameters in the LOOCV training set, covering the hyperparameter ranges shown in Table S4. For each number of features selected during the RFE step, we conducted a grid search in the hyperparameter space to optimise the area under the receiver operating characteristic curve. Once the optimal hyperparameters were found (Table S5), we determined model parameters by re-fitting the model to the training set. To increase the robustness of the model with this dataset, LOOCV was done in 20 iterations, leaving one of the NAXIVA patients out at a time. The prediction for each left-out patient was done according to the model trained on the remaining 19 patients.

**Feature importance.** We evaluated feature importance in two different steps. First, we computed the frequency with which features were selected after the recursive feature elimination. We repeated the process for each of the LOOCV iterations, which means that features could be selected between 0 and a maximum of 20 times. Figure 4e displays only features that were chosen at least eight out of twenty times in each cross-validation loop. Second, we computed the importance (i.e. weight) of each individual feature within the logistic regression

algorithm. The weights were averaged across the iterations in which they were picked.

**External validation.** No other studies have published the blood plasma markers and histopathological features following treatment with anti-angiogenic therapy in ccRCC with VTT. However, we leveraged publicly available transcriptome datasets from Phase III trials in advanced ccRCC to validate our findings.

(1) We assigned each patient in the Javelin Renal 101 transcriptomics set[16] a response and non-response pseudo-signature score using the mean expression of the genes encoding the proteins identified in the NAXIVA baseline model. For response, the genes used were *PECAM1* (encoding CD31) and *CD34* (encoding CD34). The non-response genes used were *CCL17*, *CCR4* (the CCL17 receptor, which our scRNA-seq data suggests is expressed by CD4+ T cells in tumour tissue), *IL12A* and *IL12B* (the genes encoding the IL-12p70 subunits). The patients in the top quartile for response and in the bottom quartile for non-response were included in the NAX-hi group ($n$ = 47). The patients in the bottom quartile for response and in the top quartile for non-response were included in the NAX-lo group ($n$ = 28). Survival analyses of these two groups were done using a log-rank test using the Survival (v3.7.0) and Survminer (v0.5.0) R packages.

(2) The genes in both signatures were mapped onto the differential expression data from the IMmotion151 transcriptomic dataset.

## Inclusion & ethics statement

This research included local researchers throughout the research process – study design, study implementation, data ownership, and authorship of publications. This research is relevant to a global disease. This research was approved by the local research ethics committee (REC reference: 17/EE/0240).

## Reporting summary

Further information on research design is available in the Nature Portfolio Reporting Summary linked to this article.

## Data availability

De-identified RNA-seq data have been deposited in the Gene Expression Omnibus, accession GSE281304. The de-identified imputed, normalised dataset, which was input to the ML models, is provided in the Supplementary Information file. The source data generated in this study are provided in the Source Data file. Source data are provided with this paper.

## Code availability

The data and code used to develop the baseline and dynamic machine learning models have been deposited in a GitHub repository at https://github.com/CrispinLab/NAXIVA with https://doi.org/10.5281/zenodo.14773991.

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

## Acknowledgements

The authors thank the patients and families who participated in the NAXIVA trial. We thank the CRUK Cambridge Institute genomics, bioinformatics, compliance & biobanking and histopathology Core Facilities for their technical support. We thank the Human Research Tissue Bank and The Core Biochemical Assay Laboratory at Cambridge University Hospitals NHS Foundation Trust for their technical support. We thank the NIHR Cambridge BRC Cell Phenotyping Hub and the NIHR Cambridge Biomedical Research Centre for their technical support. We thank the staff at the Scottish Clinical Trials Research Unit for their support in delivering the NAXIVA trial. NAXIVA was endorsed by Cancer Research UK (A23471). We acknowledge the support of the National Institute for Health Research Clinical Research Network (NIHR CRN). ARTIST is supported by the Cancer Research UK Cambridge Centre [CTRQQR-2021\100012]. We are grateful to V. Gnanapragasam, chief investigator of the DIAMOND study (REC 03/018), for access to renal tumour clinical samples. Infrastructure for the DIAMOND study was provided by the Cancer Research UK Cambridge Cancer Centre (Major Centre Award C9685/A25117) and the NIHR Biomedical Research Centre. This work was supported by the Cancer Research UK Cambridge Centre [CTRQQR-2021\100012; and C9685/A25117], The Mark Foundation for Cancer Research [RG95043], the Cancer Molecular Diagnostics Lab at the Cancer Research UK Cambridge Centre [CTRQQR-2021\100012], and NIHR Cambridge Biomedical Research Centre (NIHR203312). The Cambridge Human Research Tissue Bank was supported by the NIHR Cambridge Biomedical Research Centre (NIHR203312). The views expressed are those of the author(s) and not necessarily those of the NIHR or the Department of Health and Social Care. Funding and Medicine for this Investigator Sponsored Research study were provided by Pfizer Ltd. J.J. was supported by an NIHR Clinical Lectureship. G.D.S. was supported by The Mark Foundation for Cancer Research [RG95043], the Cancer Research UK Cambridge Centre [C9685/A25177 and CTRQQR-2021\100012] and NIHR Cambridge Biomedical Research Centre [NIHR203312]. M.C.O. was supported by the Joseph Mitchell Cancer Research Fund, the Academy of Medical Sciences [G117526] and NIHR [NIHR206092]. H.P. was supported by AstraZeneca UK Limited & NIHR BioResource Infrastructure Studentship (G127831). R.W. was supported by the Cancer Research UK Cambridge Centre [CTRQQR-2021\100012]. F.A.G. was supported by Cancer Research UK (C19212/A27150). J.D.S. supported by MRC [MC_UU_12022/5]. M.D.R. was supported by a Wellcome Trust Discovery Award [227432/Z/23/Z] and Cancer Research UK (A22257). A.L. has received a Career Researcher Fellowship from NHS Research Scotland to support this work.

## Author contributions

Conception and design: G.D.S., S.J.W., M.C.O. and J.O.J. Model development: I.M., H.P. and R.W. Acquisition of the data: J.O.J., S.J.W., J.B., F.E., A.E., F.A.G., I.A.M., T.J.M., L.W., A.Y.W. and S.U. Analysis and interpretation of the data: R.W., H.P., J.O.J., J.D.S., M.D.R., S.J.W., M.C.O. and G.D.S. Drafting of the manuscript: R.W., H.P., M.C.O., G.D.S. and J.O.J. Review and revision of the manuscript: R.W., H.P., I.M., J.B., F.E., A.E., F.A.G., I.A.M., T.J.M., M.D.R., J.D.S., S.U., L.W., A.Y.W., S.J.W., M.C.O., G.D.S. and J.O.J. Statistical analysis: J.O.J., R.W. and H.P. Running the clinical study: NAXIVA trial group members. Supervision: G.D.S., M.C.O. and J.O.J.

## Competing interests

The authors declare the following competing interests: H.P.: AstraZeneca (studentship). F.A.G.: research support from GE Healthcare; Grants from GSK; Consulting for AZ on behalf of the University of Cambridge. M.C.O.: 52 North Health Ltd (co-founder and employee), GE HealthCare (research funding), GSK (speaking fees). G.D.S.: Financial Interests – Evinova (consultancy, workshop on new product), British Journal of Urology International (Associate Editor) - paid role, NATCAN (Clinical Director (surgery) for the National Kidney Cancer), Audit (paid role), National Institute for Health and Care Excellence (Topic Advisor of kidney cancer guideline - paid role), AstraZeneca (Institutional Funding of the WIRE clinical trial); Non-Financial Interests - Getting It Right First Time (Principal Investigator, Chair of the kidney cancer pathway), British Association of Urological Surgeons (Member), European Association of Urology (Member); Other – MSD (Funded attendance at ESMO 2023). J.J.: Financial interests: Evinova (consultancy, workshop on new product), AstraZeneca (Institutional Funding of the WIRE clinical trial). The remaining authors declare no competing interests.

## Additional information

## the NAXIVA Study Group

Niki Couper[11], Lisa E. M. Hopcroft[12], Robert Hill[11], Athena Matakidou[4], Cara Caasi[4], James Watson[4], Ruby Cross[4], Sarah W. Burge[2,3], Anne George[4], Tobias Klatte[4,13], Tevita F. Aho[4], James N. Armitage[4], Sabrina Helena Rossi[4], Charlie Massie[2], Shubha Anand[14], Tiffany Haddow[14], Marc Dodd[14], Wenhan Deng[14], Ezequiel Martin[14], Philip Howden[15], Stephanie Wenlock[15], Evis Sala[6], Stefan Symeonides[16,17], Lynn Ho[18,19], Jennifer Baxter[17], Stuart Leslie[17], Duncan McLaren[17], John Brush[19], Marie O'Donnell[19], Alisa Griffin[20], Ruth Orr[21], Catriona Cowan[20], Thomas Powles[22], Anna Pejnovic[23], Sophia Tincey[23], Lee Grant[23], Martin Nuttall[24], Lucy Willsher[24], Christian Barnett[24], David Nicol[25], James Larkin[25], Alison Fielding[26], Christopher G. Smith[5], Axel Bex[23,27,28], Ekaterini Boleti[23], Jade Carruthers[11], Tim Eisen[2], Kate Fife[4], Angela Godoy[2], Abdel Hamid[24], Alexander Laird[19,29], Steve Leung[17], Jahangeer Malik[17], Faiz Mumtaz[23], Grenville Oades[30], Andrew N. Priest[4,6], Antony C. P. Riddick[4], Balaji Venugopal[21], Michelle Welsh[11], Kathleen Riddle[11] & Robert J. Jones[31]

[11]Scottish Clinical Trials Research Unit, Public Health Scotland, Edinburgh, UK. [12]Data & Analytics, NHS National Services Scotland, Edinburgh, UK. [13]Faculty of Health Sciences Brandenburg, Brandenburg Medical School Theodor Fontane, Brandenburg, Germany. [14]Cancer and Molecular Diagnostics Laboratory, University of Cambridge, Cambridge, UK. [15]Cambridge Genomic Services, Department of Pathology, University of Cambridge, Cambridge, UK. [16]Edinburgh Experimental Cancer Medicine Centre, University of Edinburgh, Edinburgh, UK. [17]Edinburgh Cancer Centre, NHS Lothian, Edinburgh, UK. [18]NHS Lothian, Edinburgh, UK. [19]Western General Hospital, Edinburgh, UK. [20]Beatson Cancer Centre, Glasgow, UK. [21]Beatson West of Scotland Cancer Centre, Glasgow, UK. [22]Barts Cancer Institute, London, UK. [23]Royal Free London NHS Foundation Trust, London, UK. [24]Mid and South Essex NHS Foundation Trust, Essex, UK. [25]The Royal Marsden Hospital, London, UK. [26]Patient representative, Action Kidney Cancer, Manchester, UK. [27]University College London, Division of Surgery and Interventional Science, London, UK. [28]Surgical Oncological Division, The Netherlands Cancer Institute, Amsterdam, The Netherlands. [29]Institute of Genetics and Cancer, University of Edinburgh, Edinburgh, UK. [30]Queen Elizabeth University Hospital, Glasgow, UK. [31]University of Glasgow, Glasgow, UK.

