## [Transparent Peer Review file · Nature Communications]

Angiogenic and Immune Predictors of Neoadjuvant Axitinib Response in Renal Cell Carcinoma with Venous Tumour Thrombus

Corresponding Author: Dr James Jones

Version 0:

Reviewer comments:

Reviewer #1

(Remarks to the Author)

In this study, the authors employed digital pathology, flow cytometry, plasma cytokine profiling and RNA sequencing to develop a model for predicting patient response to neoadjuvant axitinib. They found that the model achieved an AUC of 0.868 using data from week 1, and its predictive performance improved to 0.945 when using data from week 3. Overall, we found the paper to be well-written, and the findings are of interests. However, there are a few points that need to be addressed to improve the current version. We would appreciate it if authors could clarify these following questions:

- 1) Figure 3a: The dataset comprised 20 samples. Could author explain why only 6 responders and 7 non-responders are shown in the figure?
- 2) Machine learning model: Is there a specific reason for the selecting only 3 features, instead of 5, or 10, or another numbers? It would be helpful if the authors could demonstrate the results for various feature selection sizes to show the model's robustness, particularly where external data are lacking.
- 3) Model using week-3 data: Could the authors show the overlap between the features selected from week-1 data and week-3 data? Additionally, what is the model's performance if the features selected from week-1 data are fixed, with the addition of the dynamic marker assessment feature, without re-running the features selection on week-1 data?
- 4) External validation: Our primary concern with this study is the lack of external validation. Without external validation, it is difficult to assess the generalizability of the model, particularly since it was trained on only 20 samples.

(Remarks on code availability)

Reviewer #2

(Remarks to the Author)

In this manuscript, Wray et al. conducted a comprehensive multiparametric investigation of samples collected during the NAXIVA study, utilizing digital pathology, flow cytometry, plasma cytokine profiling, and RNA sequencing to assess the non-response to neoadjuvant axitinib treatments. The intriguing findings reveal that responders exhibited higher baseline microvessel density and increased induction of VEGF-A and PIGF during treatment. The distinct transcriptomic profiles observed between responders and non-responders are novel and provide a rationale for developing an ML model to predict the response to neoadjuvant axitinib treatment. Overall, the manuscript is well-written and presents novel findings. However, several concerns need to be addressed before it can be accepted for publication.

Major Comments:

1. The lack of significant differences in immune features between responders and non-responders is somewhat unexpected and may be attributed to the nature of the NAXIVA trial, which involves TKI treatments. A recent phase 2 study, NEOTAX (PMID: 39362847), which investigated neoadjuvant toripalimab plus axitinib for clear cell renal cell carcinoma with inferior vena cava tumor thrombus, should be referenced and discussed in the introduction and/or discussion sections of the manuscript. Additionally, the authors should clarify the subset of patients used in corresponding clinical trials, such as IMmotion151 and Javelin Renal 101. The response signature of NAXIVA is compared to that of TKI-treated patients but not to TKI combined with ICI (Figure 3).
2. Given that the authors claim that analyzing small clinical trial datasets may lead to overfitting of the ML model, it would be

interesting to validate the ML model in an independent dataset, such as IMmotion151 and Javelin Renal 101. Alternatively, the authors could validate a sub-model, considering that baseline blood tests and early dynamic measures may not be available in these published trials.

3. In Figure 1d and 1e, it would be beneficial to include a whole slide image as shown in Figure 1c. Furthermore, including several more representative images from different patients would provide more informative insights.

(Remarks on code availability)

Reviewer #3

(Remarks to the Author)

(Remarks on code availability)

The code is not available yet.

Reviewer #4

(Remarks to the Author)

In this manuscript Wray et al report the results of a prospective study aiming to provide a multiparametric investigation of samples collected during the NAXIVA trial (Study of Axitinib for Reducing Extent of Venous Tumour Thrombus in Renal Cancer with Venous Invasion, NCT03494816) using digital pathology, flow cytometry, plasma cytokine profiling and RNA sequencing. The authors found that responders to neoadjuvant axitinib had higher baseline microvessel density and increased induction of VEGF-A and PIGF during treatment. A multimodal machine learning model integrating features predicted response with an AUC of 0.868, improving to 0.945 when using features from week 3. Key predictive features included plasma CCL17 and IL-12.

The authors should be congratulated for their elegant, comprehensive and robust analysis using data from a pivotal phase II neoadjuvant trial in the field of RCC.

The manuscript is well written, properly organized (nice integration between text, tables, and figures), and extremely well-readable, even by clinicians with little knowledge on translational research.

The findings of the study are original and represent meaningful data that add to the current knowledge from both clinical and research perspectives. The figures are just great and comprehensive.

I do not have any major concern related to the study concept, design, and analysis.

The authors may further improve the manuscript by address the following points:

1. A Figure/Table summarizing the study findings (i.e. all putative predictive biomarkers of response to neoadjuvant axitinib) in the form of a "graphical abstract" could serve readers who are less experienced in translational research and may enhance the impact of the study
2. Regarding the machine learning model integrating multiple baseline features predicts treatment response: the definition of response is not necessarily clinically meaningful (response is defined as a >30% reduction in VTT length compared to baseline). Could the authors expand the definition on how such a definition of response translates into harder clinical endpoints in the study (any change in surgical approach, patient outcome or surgery-related variable?). For the ML model, could the authors evaluate other endpoints? (e.g. change in surgical strategy, reduction of surgery's invasiveness, RFS, etc.)?
3. Assessing the specific analyses on the ML model to predict response is beyond this Reviewer's knowledge and should be extensively checked by an expert biostatistician.

(Remarks on code availability)

Reviewer #5

(Remarks to the Author)

The paper titled "Angiogenic and Immune Predictors of Neoadjuvant Axitinib Response in Renal Cell Carcinoma with Venous Tumour Thrombus" presents an in-depth exploration of predictors for treatment response in patients with clear cell renal cell carcinoma (ccRCC) with venous tumour thrombus (VTT) undergoing neoadjuvant axitinib therapy.

The key results include:

- Higher baseline microvessel density (MVD) in responders to axitinib, which was significantly associated with a better treatment response.
- Circulating angiogenic factors such as VEGF-A and PIGF were differentially induced in responders, with a notable 7-fold increase in PIGF in responders by week 3.
- Immune shifts were observed, with non-responders showing higher levels of immune cytokines IL-12p70 and IL-7, indicating a possible immune suppression in these patients.
- The machine learning model developed in the study successfully predicted treatment response, with high AUC (0.868 at

baseline, 0.945 after three weeks), identifying key biomarkers such as CCL17 and IL-12p70. This work provides novel insights into the predictive markers of VTT response to neoadjuvant axitinib, particularly in terms of angiogenic and immune features. It builds on existing literature regarding angiogenesis in ccRCC (e.g., VEGF, PIGF) but offers new findings in relation to immune profiles and their predictive value. While there are other studies exploring VEGF-related biomarkers in RCC, this study's combination of tissue, blood, and machine learning-based data is innovative. The originality lies in integrating angiogenic and immune markers alongside RNA sequencing data, enabling a personalized approach to therapy.

It is consistent with previous work, such as the studies from McDermott et al. (2018) and Motzer et al. (2020), which discuss the use of immunotherapy and VEGFR-TKIs in RCC, particularly with respect to angiogenesis. The novel use of machine learning to combine these features provides a new approach to predicting treatment response in clinical settings. The work supports its conclusions that angiogenic markers (e.g., PIGF, VEGF-A) and immune-related factors (e.g., IL-12p70, CCL17) are key predictors of treatment response to axitinib in VTT. However, due to the small sample size (20 patients), further validation in larger cohorts is needed to confirm the generalizability of these findings. Additionally, external validation of the machine learning model would be critical to establishing its predictive power in broader clinical settings.

There are no major flaws in the data analysis. The methods used for tissue and immune profiling (e.g., digital pathology, RNA sequencing, flow cytometry) are robust. The statistical analyses are appropriate, with the use of cross-validation in the machine learning model reducing the risk of overfitting.

The methodology is sound and aligns with the high standards expected in oncology and molecular medicine. The multiparametric approach integrating imaging, blood biomarkers, tissue profiling, RNA sequencing, and machine learning is a comprehensive method for identifying response predictors in this complex clinical scenario. This approach is novel and well-suited to the multifactorial nature of RCC with VTT.

The methods are detailed enough for the study to be reproducible. The paper provides clear descriptions of the experimental procedures used in the analysis (e.g., flow cytometry, RNA-seq, image analysis). Additionally, the machine learning model is thoroughly explained, including data pre-processing, feature selection, and cross-validation approach. The availability of code and de-identified data upon acceptance further enhances the reproducibility of the study.

However, there are limitations:

- Sample size: With only 20 patients, the study's findings are based on a small dataset, and the predictive power of the machine learning model could be compromised by this.
- Absence of external validation: The model's generalizability would be improved with validation on an independent cohort.
- Lack of functional validation: The study relies on correlative biomarkers without experimental validation of their causal role in response to treatment.

These limitations do not prohibit publication but should be addressed in future studies.

Final Recommendations:

This study provides valuable insights into predictive biomarkers for treatment response in ccRCC with VTT undergoing neoadjuvant axitinib. However, further validation in larger cohorts and external datasets is necessary to confirm the findings. The manuscript is well-written and provides significant contributions to the field, particularly in terms of combining angiogenic and immune features for prediction models. Given the high-quality methodology and the potential clinical relevance of the findings, I would recommend publication with minor revisions, especially concerning the external validation and sample size limitations.

(Remarks on code availability)

Version 1:

Reviewer comments:

Reviewer #1

(Remarks to the Author)

The authors have adequately addressed our comments, and we have no additional questions.

(Remarks on code availability)

Reviewer #2

(Remarks to the Author)

All my comments have been addressed.

(Remarks on code availability)

Reviewer #3

(Remarks to the Author)

I co-reviewed this manuscript with one of the reviewers who provided the listed reports. This is part of the Nature Communications initiative to facilitate training in peer review and to provide appropriate recognition for Early Career

Researchers who co-review manuscripts.

(Remarks on code availability)

Reviewer #4

(Remarks to the Author)

The authors have revised the manuscript according to the Reviewers' comments and suggestions, significantly improving the quality of the manuscript and the strengths of the analyses. Moreover, they provided extensive, transparent explanations for all changes in the manuscript and for all aspects of concern raised by the reviewers.

I congratulate the authors for their endeavours and for the quality of their study.

(Remarks on code availability)

Reviewer #5

(Remarks to the Author)

The authors have effectively addressed reviewer comments, strengthening the manuscript with key revisions. They emphasized the use of leave-one-out cross-validation to mitigate the challenges of a small sample size and demonstrated that the predictive model is robust despite limited data.

The manuscript now includes external validation using transcriptomic data from the IMmotion151 and JAVELIN Renal 101 trials, supporting the model's predictive power. The authors also acknowledged the need for functional validation, outlining future steps in subsequent studies.

With these revisions, the paper provides solid evidence for predictive biomarkers in RCC with VTT, showcasing the potential of multiparametric approaches in personalized oncology. The study is now ready for publication in Nature Communications.

(Remarks on code availability)

Reviewer 1

Comment 1

Figure 3a: The dataset comprised 20 samples. Could author explain why only 6 responders and 7 non-responders are shown in the figure?

Response:

Thank you for highlighting this. We extracted sufficient RNA for sequencing from 16 baseline biopsies of the 20 patients. After sequencing, 3 of these were not high enough quality to be included in the differential expression analysis.

Changes implemented:

We have included a comment to explain this in the RNA methods and results.

In Methods, section **RNA-seq** (new text highlighted in blue)

“RNA extraction & sequencing: RNA was extracted from formalin-fixed paraffin-embedded (FFPE) tissue using the ReliaPrep FFPE Total RNA kit, according to manufacturer’s instructions, and assessed by Qubit and Agilent RNA ScreenTape System and for quantity and quality. We extracted sufficient RNA from 16 of the 20 samples.”

In Results, section **Responders and non-responders have distinct transcriptomic profiles**

“RNA-seq was performed on pre-treatment biopsies to investigate transcriptomic predictors of response, with 13 biopsies meeting our quality criteria (see Methods).”

Comment 2

Machine learning model: Is there a specific reason for the selecting only 3 features, instead of 5, or 10, or another numbers? It would be helpful if the authors could demonstrate the results for various feature selection sizes to show the model’s robustness, particularly where external data are lacking.

Response:

The number of features is the result of the model optimization process. We tested feature selection up to 7 features within a cross-validation setting, and the performance was best for 3 features in both models.

Changes implemented:

We have added Table S1 summarising the performance of these models and the features selected.

In Methods, section **Machine learning models**

“We used a five-fold cross validation setup to optimise model hyperparameters in the LOOCV training set, covering the hyperparameter ranges shown in Table S4. For each number of features selected during the RFE step, we conducted a grid search in the hyperparameter space to optimise the area under the receiver operating characteristic

curve.”

In Results, section ***A machine learning model integrating multiple baseline features predicts treatment response***

“In this model (“baseline model”), best performance was achieved when three features of the 62 were selected (Table S1).”

In Supplementary Material

Table S1. AUC and feature selection results for machine learning models with different numbers of features pre-specified for selection by the Recursive Feature Elimination (RFE) algorithm. Only the top N selected features are listed for every “N features”, based on their selection frequency. The approaches include:

- Selection from baseline features only (this approach with N=3 selected features was further analysed as the ‘baseline model’).
- Selection from baseline and dynamic week 3 features (this approach with N=3 selected features was analysed as the ‘dynamic model’).
- Selection from dynamic week 3 features, concatenated with a fixed set of 3 baseline features (CCL17, IL12-p70, and CD31+ MVD, manually pre-specified based on results from the baseline model).

N features	Baseline only	Baseline + Dynamic	Fixed Baseline + Model-selected Dynamic
2	AUC = 0.802 Baseline: CCL17, IL-12p70	AUC = 0.670 Baseline: CCL17 Dynamic: PIGF	N/A
3	AUC = 0.868 Baseline: CCL17, IL-12p70, CD31+ MVD	AUC = 0.945 Baseline: CCL17 Dynamic: PIGF, Tie-2	N/A
4	AUC = 0.758 Baseline: CCL17, IL-12p70, CD31+ MVD, sVEGFR-1	AUC = 0.593 Baseline: CCL17, IL-12p70 Dynamic: PIGF, Tie-2	AUC = 0.769 Baseline (fixed): CCL17, IL-12p70, CD31+ MVD Dynamic: PIGF
5	AUC = 0.802 Baseline: CCL17, IL-12p70, CD31+ MVD, sVEGFR-1, LTa	AUC = 0.780 Baseline: CCL17, IL-12p70, sVEGFR-1 Dynamic: PIGF, Tie-2	AUC = 0.813 Baseline (fixed): CCL17, IL-12p70, CD31+ MVD Dynamic: PIGF, Tie-2
6	AUC = 0.813 Baseline: CCL17, IL-12p70, CD31+ MVD, sVEGFR-1, LTa, FGF	AUC = 0.868 Baseline: CCL17, IL-12p70, sVEGFR-1, FGF Dynamic: PIGF, Tie-2	AUC = 0.934 Baseline (fixed): CCL17, IL-12p70, CD31+ MVD Dynamic: PIGF, Tie-2, FGF
7	AUC = 0.714 Baseline: CCL17, IL-12p70, CD31+ MVD, sVEGFR-1, LTa, FGF, IL-7	AUC = 0.8791 Baseline: CCL17, IL-12p70, sVEGFR-1, FGF Dynamic: PIGF, Tie-2, FGF	AUC = 0.934 Baseline (fixed): CCL17, IL-12p70, CD31+ MVD Dynamic: PIGF, Tie-2, FGF, VEGF-A

Comment 3a

Model using week-3 data: Could the authors show the overlap between the features selected from week-1 data and week-3 data?

Response:

This is a good suggestion, thank you.

Changes implemented:

We have adjusted Table 2 to show which features are selected in each model.

Table 2: Most significant features identified in the ML models

Marker	Sample	Timepoint	Selected in baseline model	Selected in dynamic model
CD31	Tissue	Week 1	* -	
CD34	Tissue	Week 1	* -	
IL-12p70	Blood	Week 1	* -	* -
CCL17	Blood	Week 1	* -	* -
PIGF	Blood	Week 3		* -
sTie-2	Blood	Week 3		* -

Comment 3b

Additionally, what is the model's performance if the features selected from week-1 data are fixed, with the addition of the dynamic marker assessment feature, without re-running the features selection on week-1 data?

Response:

Thank you for this suggestion - we have developed the model suggested by the reviewer. The input to this new model consists of three fixed baseline features (CCL17, IL-12p70, and CD31+ MVD (based on results from the baseline model) and a pre-set number of week-3 fold-change features selected with recursive feature elimination (RFE) during leave-one-out iterations. We varied the pre-set number of RFE-selected week-3 features between one and four, each time training a new version of the model, with the total number of input features varying between four and seven (three fixed features plus 1-4 dynamic features).

With this approach, the best performance was observed with six and seven features, reaching AUC of 0.934. This is better than performance of the baseline model (AUC = 0.868), but slightly worse than the performance of the dynamic model (AUC = 0.945), where a total of three features are RFE-selected from both week-1 and week-3 fold-change datasets.

Changes implemented:

We have included the findings from the model using this alternative approach in the supplement.

In Materials and Methods, section **Machine learning models:**

“The models included recursive feature elimination (RFE) using a logistic regression estimator; the number of features selected by RFE was varied empirically, and several

approaches of combining baseline and week 3 fold-change features were explored; the method achieving the highest AUC was selected (N=3 features, Table S1). The predictions were done by logistic regression with stochastic gradient descent, all coded in Python using scikit-learn version 1.4.

In Supplementary Material:

“Table S1. AUC and feature selection results for machine learning models with different numbers of features pre-specified for selection by the Recursive Feature Elimination (RFE) algorithm. Only the top N selected features are listed for every “N features”, based on their selection frequency. The approaches include:

- Selection from baseline features only (this approach with N=3 selected features was further analysed as the ‘baseline model’),
- Selection from baseline and dynamic week 3 features (this approach with N=3 selected features was analysed as the ‘dynamic model’),
- Selection from dynamic week 3 features, concatenated with a fixed set of 3 baseline features (CCL17, IL12-p70, and CD31+ MVD, manually pre-specified based on results from the baseline model).”

N features	Baseline only	Baseline + Dynamic	Fixed Baseline + Model-selected Dynamic
2	AUC = 0.802 Baseline: CCL17, IL-12p70	AUC = 0.670 Baseline: CCL17 Dynamic: PIGF	N/A
3	AUC = 0.868 Baseline: CCL17, IL-12p70, CD31+ MVD	AUC = 0.945 Baseline: CCL17 Dynamic: PIGF, Tie-2	N/A
4	AUC = 0.758 Baseline: CCL17, IL-12p70, CD31+ MVD, sVEGFR-1	AUC = 0.593 Baseline: CCL17, IL-12p70 Dynamic: PIGF, Tie-2	AUC = 0.769 Baseline (fixed): CCL17, IL-12p70, CD31+ MVD Dynamic: PIGF
5	AUC = 0.802 Baseline: CCL17, IL-12p70, CD31+ MVD, sVEGFR-1, LTa	AUC = 0.780 Baseline: CCL17, IL-12p70, sVEGFR-1 Dynamic: PIGF, Tie-2	AUC = 0.813 Baseline (fixed): CCL17, IL-12p70, CD31+ MVD Dynamic: PIGF, Tie-2
6	AUC = 0.813 Baseline: CCL17, IL-12p70, CD31+ MVD, sVEGFR-1, LTa, FGF	AUC = 0.868 Baseline: CCL17, IL-12p70, sVEGFR-1, FGF Dynamic: PIGF, Tie-2	AUC = 0.934 Baseline (fixed): CCL17, IL-12p70, CD31+ MVD Dynamic: PIGF, Tie-2, FGF
7	AUC = 0.714 Baseline: CCL17, IL-12p70, CD31+ MVD, sVEGFR-1, LTa, FGF, IL-7	AUC = 0.8791 Baseline: CCL17, IL-12p70, sVEGFR-1, FGF Dynamic: PIGF, Tie-2, FGF	AUC = 0.934 Baseline (fixed): CCL17, IL-12p70, CD31+ MVD Dynamic: PIGF, Tie-2, FGF, VEGF-A

Comment 4

External validation: Our primary concern with this study is the lack of external validation. Without external validation, it is difficult to assess the generalizability of the model, particularly since it was trained on only 20 samples.

Response:

Thank you for making this important point about validation.

Firstly, we would like to point out that the model was trained using a strict leave-one-out cross validation approach (with nested parameter optimisation to avoid any leakage), which is a statistically robust approach for building models on small datasets and testing their generalizability.

Secondly, to the reviewer's point – we agree that it would be ideal to have an equivalent external dataset; however, we are not aware of any other clinical studies or trials with equivalent data. Future validation work will involve an ongoing clinical trial led by our team, WIRE (NCT03741426), a novel multi-centre trial which will measure the same markers as tested in NAXIVA. We anticipate that WIRE will serve as future validation for the role of CCL17, IL-12p70, and the dynamic markers PIGF and sTie-2. However, WIRE is expected to continue to recruit patients for several years, so it is out of scope for this paper.

Instead, we have addressed the reviewer's comment by conducting an in-depth analysis of two existing trials which administer an anti-angiogenic / TKI monotherapy, and have publically available molecular data. The trials we have considered for external validation are in the table below.

Trial	Feature type	Primary endpoint	Drug	Patient population	PMID
JAVELIN Renal 101 (NCT02684006)	Transcriptomics	Progression-free survival	Sunitinib, Avelumab + Axitinib	Advanced RCC	32895571 ¹
IMmotion151 (NCT02420821)	Transcriptomics	Progression-free survival NB: PFS data not published	Sunitinib, Atezolizumab + Bevacizumab	Advanced RCC	33157048 ²

Both trials measure tumour transcriptomics. To validate our results, we attempted to find relationships between the expression of the genes encoding the plasma proteins we identified and treatment response in these trials.

For IMmotion151, we plotted the genes encoding our proteins of interest on the published differential gene expression results from the trial (Fig. 4g).

This differential gene expression is done on a cluster-vs-everything-else basis from the transcriptomes of the pre-treatment tumours in the 823 patients. We assume that the patients with the same inclusion profile as NAXIVA (i.e. ccRCC with VTT) are represented in the large IMmotion151 study of advanced kidney cancer patients (10-15% patients have a VTT^{3,4}). The markers identified by the baseline and dynamic models point towards our responders belonging to clusters 1 or 2, which the IMmotion151 translational study found related to the response of the patients to anti-angiogenic therapy. The CCL17 receptor, CCR4, is over-represented in the IMmotion151 patients belonging to cluster 4, which the

IMmotion151 study identified to be related to immunotherapy response. While these relationships do not fully validate our results, we are encouraged that this large, transcriptomic dataset is able to represent our plasma results to a certain degree.

For JAVELIN Renal 101, we have conducted further analysis, using the published RNA count data and progression-free survival data. We calculated a 'pseudo-signature' score using the mean expression of the genes identified in the NAXIVA baseline model for response: *PECAM1* (encoding CD31), *CD34* (encoding CD34); and non-response: *CCL17*, *CCR4* (*CCL17* receptor), *IL12A* and *IL12B*. We included *CCR4* in the non-response signature because our single-cell RNA-seq data suggests that there is low expression of *CCL17* in ccRCC tumours and associated tissues, whereas *CCR4* is expressed by CD4+ T cells in the tumour (Fig. S10a). The patients with a response signature score in the top quartile and with a non-response signature score in the bottom quartile were designated 'Nax-hi'. The patients with a response signature in the bottom quartile and a non-response signature in the top quartile were designated 'Nax-lo'. The survival of both of these groups was plotted.

We find that there is a significant difference between the patients in our Nax-hi group and in the Nax-lo group in the sunitinib arm of the JAVELIN trial. However, the number of patients in each group is low. We acknowledge that this dataset is not fully comparable to NAXIVA because progression-free survival is mechanistically different to our measure for outcome in the NAXIVA trial (VTT length change), and because the patients in the JAVELIN dataset all have confirmed advanced or metastatic ccRCC, whereas the NAXIVA patients have either M0 (localised) or M1 disease (Fig. S1f, reproduced below). However, our data suggests that there is a survival difference between patients whose tumours express the genes which our baseline model highlighted as important.

We also stratified the JAVELIN patients using the top differentially expressed genes in NAXIVA ($p < 0.05$, as in Fig. 3b), this time where the 'response' signature included the

genes with LFC > 0 and 'non-response' with LFC < 0. The patients with a response signature score in the top quartile and with a non-response signature score in the bottom quartile were designated 'Nax-RNA-hi'. The patients with a response signature score in the bottom quartile and a non-response signature score were designated 'Nax-RNA-lo'.

This RNA-based signature is able to stratify the patients in the sunitinib arm of the trial.

In summary, we are encouraged by the mapping of our markers to the IMmotion151 transcriptomic data which indicates that the expression of some of the plasma features and their receptors may be upregulated in the tumours of patients most likely to respond to anti-angiogenic therapy. We find that the patients in the sunitinib arm of the JAVELIN Renal 101 study can be meaningfully separated based on the genes encoding our proteins of interest, and by the genes most differentially expressed by the NAXIVA responders and non-responders. Despite the size of these studies, they are not easily comparable to NAXIVA in terms of data type, clinical endpoint or patient population. Our analysis of existing single-cell transcriptome data suggests to us that CCL17 is produced in tissues other than the tumour site, which makes the transcriptomic markers measured in these datasets difficult to compare to our study. We suspect that analysis of the plasma proteome offers alternative insight to response to the tumour to transcriptomic data. We hope that the sharing of this data allows further validation in the future, including the data from the WIRE trial that is currently being collected.

Changes implemented:

We have included these graphs in the main figures and supplementary figures, and edited our text to explain the methods for external validation.

In Methods, **Machine learning models**

“External validation: No other studies have published the blood plasma markers and histopathological features following treatment with anti-angiogenic therapy in ccRCC with VTT. However, we leveraged publicly available transcriptome datasets from Phase III trials in advanced ccRCC to validate our findings.”

(1) We assigned each patient in the Javelin Renal 101 transcriptomics set¹⁵ a ‘response’ and ‘non-response’ pseudo-signature score using the mean expression of the genes encoding the proteins identified in the NAXIVA baseline model. For response, the genes used were *PECAM1* (encoding CD31) and *CD34* (encoding CD34). The non-response genes used were *CCL17*, *CCR4* (the *CCL17* receptor, which our scRNA-seq data suggests is expressed by CD4+ T cells in tumour tissue, Fig. S10a), *IL12A* and *IL12B* (the genes encoding the IL-12p70 subunits). The patients in the top quartile for response and in the bottom quartile for non-response were included in the ‘NAX-hi’ group (n=47). The

patients in the bottom quartile for response and in the top quartile for non-response were included in the 'NAX-lo' group (n=28). Survival analyses of these two groups was done using a log rank test using the Survival (v3.7.0) and Survminer (v0.5.0) R packages. (2) The genes in both signatures were mapped onto the differential expression data from the IMmotion151 transcriptomic dataset¹⁷.”

In Results, **A machine learning model integrating multiple baseline features predicts treatment response**

“The expression of the genes encoding these baseline features (*PECAM1*, *CD34*, *CCL17*, *CCR4*, *IL12A*, *IL12B*) identified by the model are able to stratify the patients in Javelin Renal 101¹⁵ according to progression-free survival (log-rank test, $p = 0.034$, Fig. S11a).”

Figure S11

a, The NAXIVA baseline signature can stratify patients in the sunitinib arm of the Javelin Renal 101 trial [15].

In Methods, **RNA-seq**

“RNA signature scores: The NAX-RNA scores were generated using the genes satisfying $\text{Padj} < 0.05$ and absolute $\text{LFC} > 2$ in the NAXIVA transcriptome data. For each patient, a ‘response’ signature was calculated as the mean expression of the genes where $\text{NAXIVA LFC} > 2$, and a ‘non-response’ signature where $\text{LFC} < -2$. Javelin¹⁶ patients with a ‘response’ signature in the top quartile and a ‘non-response’ signature in the bottom quartile were termed ‘NAX-RNA-hi’, and patients with ‘response’ in the bottom quartile and ‘non-response’ in the top quartile were termed ‘NAX-RNA-lo’. Median PFS was compared between the two groups, and a log-rank test was run using Survival (v3.7.0) and Survminer (v0.5.0) R packages.”

In Results, **Responders and non-responders have distinct transcriptomic profiles**

“We generated signatures from the top differentially expressed genes in the NAXIVA patients. We applied these signatures to published patient level RNA-seq data from the Javelin Renal 101 study¹⁶, a randomised phase III clinical trial comparing axitinib plus avelumab with sunitinib in advanced renal cancer. We stratified the patients into NAX-RNA-hi and NAX-RNA-lo groups. Progression-free survival (PFS) was compared between patients in the NAX-RNA-hi and NAX-RNA-lo groups in each arm of the Javelin trial (Fig. 3d-e). For the sunitinib arm, the patients in the NAX-RNA-hi group had a higher PFS than

those in the NAX-RNA-lo group (Fig. 3d). However, this effect was not true in the avelumab+axitinib arm of the trial (Fig. 3e).”

Figure 3

d-e, The most differentially expressed genes in NAXIVA ($p < 0.05$) stratify patients according to PFS in Javelin Renal 101 study for the Sunitinib arm (**d**) and not for the avelumab+axitinib arm (**e**).

In Discussion, paragraph 4

“Our study finds that the expression of highly upregulated proteins in NAXIVA responders were also upregulated in C1/2 patients (angiogenic/angio-stromal) in publicly available IMmotion151 transcriptomic data, whereas the opposite was observed for C4/5/6 (T-effector/proliferative, proliferative and stromal/proliferative), which fitted better with the non-responding patients. We also find that the genes encoding our blood markers are able to stratify the patients in the Javelin Renal 101 trial according to progression-free survival.”

Reviewer 2

Comment 1a

The lack of significant differences in immune features between responders and non-responders is somewhat unexpected and may be attributed to the nature of the NAXIVA trial, which involves TKI treatments. A recent phase 2 study, NEOTAX (PMID: 39362847), which investigated neoadjuvant toripalimab plus axitinib for clear cell renal cell carcinoma with inferior vena cava tumor thrombus, should be referenced and discussed in the introduction and/or discussion sections of the manuscript.

Response:

Thank you for this point about referencing the recent NEOTAX study - we agree. NEOTAX tests neoadjuvant toripalimab plus axitinib and finds that pre-treatment responder biopsies have reduced T-helper cells and that pre-treatment non-responder biopsies have higher PD-1+ cytotoxic T cell infiltration. As the reviewer states, we expect that a trial administering immunotherapy would find differences in the immune cell compartment that are not found in very similar trial testing TKI-only. We also suspect that our CCL17 finding somewhat corroborates the NEOTAX findings; the predominant cell type expressing the CCL17 receptor, CCR4, are CD4+ T cells (see our single-cell RNA-seq analysis in Fig. S10a, reproduced below).

Changes implemented:

We have added a comment about the NEOTAX trial in the introduction and discussion.

In Introduction, paragraph 3 (new text highlighted in blue)

“Another recent study has shown that response to combination therapy in a VTT setting is increased when there are reduced T-helper cells in the pre-treatment tumour (NEOTAX, ChiCTR2000030405).”

In Discussion, paragraph 6

“The recent NEOTAX study finds that responders to neoadjuvant toripalimab plus axitinib

for ccRCC patients with VTT have lower densities of T helper cells in the tumor biopsy³⁸. CD4+ T cells are the predominant cell type expressing the CCL17 receptor, CCR4 (Fig. S10a). However, in our data, we did not find any clear evidence of axitinib altering the immune profile and the overall immune phenotype remained stable on treatment. This is consistent with a study of neoadjuvant pazopanib in localised RCC, which did not find any change in immune signatures on treatment²⁰. Axitinib has a narrow range of targets compared to other TKIs used in RCC³⁹, so these observations do not rule out an immune modulatory effect of TKIs that target a wider range of receptors such as lenvatinib or cabozantinib.”

Comment 1b

Additionally, the authors should clarify the subset of patients used in corresponding clinical trials, such as IMmotion151 and Javelin Renal 101. The response signature of NAXIVA is compared to that of TKI-treated patients but not to TKI combined with ICI (Figure 3).

Response:

Thank you, we agree this needs a clear explanation.

The IMmotion151 signatures are generated using non-negative matrix factorisation of the pre-treatment transcriptomes for all 823 patients². The clusters of patients are not determined by the response to therapy, but are linked to response to therapy after the clustering has been performed. We have identified our response markers in the IMmotion151 pre-treatment transcriptomic data to understand where a tumour over-expressing our proteins of interest would fall in terms of the IMmotion151 clusters. We find that the cluster into which our response signature falls is able to reflect response to anti-angiogenic therapy, which is one of our attempts to validate our findings on an external dataset.

The authors of the JAVELIN Renal 101 translational paper generated angio and immune signatures according to the pre-treatment transcriptomes of patients that responded best to sunitinib and avelumab+axitinib, respectively¹. These are used to assess whether the JAVELIN signatures are able to stratify the NAXIVA responders and non-responders into the drug group to which they are more likely to respond.

Changes implemented:

We have added comments to explain that these signatures from IMmotion151 and JAVELIN are derived from pre-treatment samples in the Results section.

In Results, section ***Responders and non-responders have distinct transcriptomic profiles***

“The most differentially expressed genes from NAXIVA were mapped onto publicly available data from a Phase III study, IMmotion151, which described seven distinct molecular clusters derived from pre-treatment tumour transcriptomes in advanced RCC¹⁷.”

“Three of the non-responders score highly in the Javelin “Immuno” score, but the spread of the scores is broad (Fig. 3f), which we expect since the Javelin Immuno score is derived from patients treated with immunotherapy.”

Comment 2

Given that the authors claim that analyzing small clinical trial datasets may lead to overfitting of the ML model, it would be interesting to validate the ML model in an independent dataset, such as IMmotion151 and Javelin Renal 101. Alternatively, the authors could validate a sub-model, considering that baseline blood tests and early dynamic measures may not be available in these published trials.

Response:

We have followed the reviewer's suggestion and have attempted to validate a 'sub-model', where the gene expression of our protein markers in the tumour is used as an imperfect proxy for their plasma concentration. The results are encouraging and suggest that there is a strong signal behind the machine learning signature.

We discussed the new results in detail as part of Reviewer 1, Comment 4 and we are including them here again to facilitate reading.

Firstly, we would like to point out that the model was trained using a strict leave-one-out cross validation approach (with nested parameter optimisation to avoid any leakage), which is a statistically robust approach for building models on small datasets and testing their generalizability.

Secondly, to the reviewer's point – we agree that it would be ideal to have an equivalent external dataset; however, we are not aware of any other clinical studies or trials with equivalent data. Future validation work will involve an ongoing clinical trial led by our team, WIRE (NCT03741426), a novel multi-centre trial which will measure the same markers as tested in NAXIVA. We anticipate that WIRE will serve as future validation for the role of CCL17, IL-12p70, and the dynamic markers PIGF and sTie-2. However, WIRE is expected to continue to recruit patients for several years, so it is out of scope for this paper.

Instead, we have addressed the reviewer's comment by conducting an in-depth analysis of two existing trials which administer an anti-angiogenic / TKI monotherapy, and have publically available molecular data. The trials we have considered for external validation are in the table below.

Trial	Feature type	Primary endpoint	Drug	Patient population	PMID
JAVELIN Renal 101 (NCT02684006)	Transcriptomics	Progression-free survival	Sunitinib, Avelumab + Axitinib	Advanced RCC	32895571 ¹
IMmotion151 (NCT02420821)	Transcriptomics	Progression-free survival NB: PFS data not published	Sunitinib, Atezolizumab + Bevacizumab	Advanced RCC	33157048 ²

Both trials measure tumour transcriptomics. To validate our results, we attempted to find relationships between the expression of the genes encoding the plasma proteins we identified and treatment response in these trials.

For IMmotion151, we plotted the genes encoding our proteins of interest on the published differential gene expression results from the trial (Fig. 4g).

This differential gene expression is done on a cluster-vs-everything-else basis from the transcriptomes of the pre-treatment tumours in the 823 patients. We assume that the patients with the same inclusion profile as NAXIVA (i.e. ccRCC with VTT) are represented in the large IMmotion151 study of advanced kidney cancer patients (10-15% patients have a VTT^{3,4}). The markers identified by the baseline and dynamic models point towards our responders belonging to clusters 1 or 2, which the IMmotion151 translational study found related to the response of the patients to anti-angiogenic therapy. The CCL17 receptor, CCR4, is over-represented in the IMmotion151 patients belonging to cluster 4, which the IMmotion151 study identified to be related to immunotherapy response. While these relationships do not fully validate our results, we are encouraged that this large, transcriptomic dataset is able to represent our plasma results to a certain degree.

For JAVELIN Renal 101, we have conducted further analysis, using the published RNA count data and progression-free survival data. We calculated a 'pseudo-signature' score using the mean expression of the genes identified in the NAXIVA baseline model for response: *PECAM1* (encoding CD31), *CD34* (encoding CD34); and non-response: *CCL17*, *CCR4* (*CCL17* receptor), *IL12A* and *IL12B*. We included *CCR4* in the non-response signature because our single-cell RNA-seq data suggests that there is low expression of *CCL17* in ccRCC tumours and associated tissues, whereas *CCR4* is expressed by CD4+ T cells in the tumour (Fig. S10a). The patients with a response signature score in the top quartile and with a non-response signature score in the bottom quartile were designated 'Nax-hi'. The patients with a response signature in the bottom quartile and a non-response signature in the top quartile were designated 'Nax-lo'. The survival of both of these groups was plotted.

We find that there is a significant difference between the patients in our Nax-hi group and in the Nax-lo group in the sunitinib arm of the JAVELIN trial. However, the number of patients in each group is low. We acknowledge that this dataset is not fully comparable to NAXIVA because progression-free survival is mechanistically different to our measure for outcome in the NAXIVA trial (VTT length change), and because the patients in the

JAVELIN dataset all have confirmed advanced or metastatic ccRCC, whereas the NAXIVA patients have either M0 (localised) or M1 disease (Fig. S1f, reproduced below). However, our data suggests that there is a survival difference between patients whose tumours express the genes which our baseline model highlighted as important.

We also stratified the JAVELIN patients using the top differentially expressed genes in NAXIVA ($p < 0.05$, as in Fig. 3b), this time where the ‘response’ signature included the genes with $LFC > 0$ and ‘non-response’ with $LFC < 0$. The patients with a response signature score in the top quartile and with a non-response signature score in the bottom quartile were designated ‘Nax-RNA-hi’. The patients with a response signature score in the bottom quartile and a non-response signature score were designated ‘Nax-RNA-lo’.

This RNA-based signature is able to stratify the patients in the sunitinib arm of the trial.

In summary, we are encouraged by the mapping of our markers to the IMmotion151 transcriptomic data which indicates that the expression of some of the plasma features and their receptors may be upregulated in the tumours of patients most likely to respond to anti-angiogenic therapy. We find that the patients in the sunitinib arm of the JAVELIN Renal 101 study can be meaningfully separated based on the genes encoding our proteins of interest, and by the genes most differentially expressed by the NAXIVA responders and non-responders. Despite the size of these studies, they are not easily comparable to NAXIVA in terms of data type, clinical endpoint or patient population. Our analysis of existing single-cell transcriptome data suggests to us that CCL17 is produced in tissues other than the tumour site, which makes the transcriptomic markers measured in these datasets difficult to compare to our study. We suspect that analysis of the plasma proteome offers alternative insight to response to the tumour to transcriptomic data. We hope that the sharing of this data allows further validation in the future, including the data from the WIRE trial that is currently being collected.

Changes implemented:

We have included these graphs in the main figures and supplementary figures, and edited our text to explain the methods for external validation.

In Methods, ***Machine learning models***

“External validation: No other studies have published the blood plasma markers and histopathological features following treatment with anti-angiogenic therapy in ccRCC with VTT. However, we leveraged publicly available transcriptome datasets from Phase III trials in advanced ccRCC to validate our findings.

(1) We assigned each patient in the Javelin Renal 101 transcriptomics set¹⁶ a ‘response’ and ‘non-response’ pseudo-signature score using the mean expression of the genes encoding the proteins identified in the NAXIVA baseline model. For response, the genes used were *PECAM1* (encoding CD31) and *CD34* (encoding CD34). The non-response genes used were *CCL17*, *CCR4* (the *CCL17* receptor, which our scRNA-seq data suggests is expressed by CD4+ T cells in tumour tissue, Fig. S10a), *IL12A* and *IL12B* (the genes encoding the IL-12p70 subunits). The patients in the top quartile for response and in the bottom quartile for non-response were included in the ‘NAX-hi’ group (n=47). The patients in the bottom quartile for response and in the top quartile for non-response were included in the ‘NAX-lo’ group (n=28). Survival analyses of these two groups was done using a log-rank test using the Survival (v3.7.0) and Survminer (v0.5.0) R packages.
(2) The genes in both signatures were mapped onto the differential expression data from the IMmotion151 transcriptomic dataset¹⁷.”

In Results, ***A machine learning model integrating multiple baseline features predicts treatment response***

“The expression of the genes encoding these baseline features (*PECAM1*, *CD34*, *CCL17*, *CCR4*, *IL12A*, *IL12B*) identified by the model are able to stratify the patients in Javelin Renal 101¹⁶ according to progression-free survival (log-rank test, p = 0.034, Fig. S11a).”

In Methods, ***RNA-seq***

“RNA signature scores: The NAX-RNA scores were generated using the genes satisfying $\text{Padj} < 0.05$ and absolute $\text{LFC} > 2$ in the NAXIVA transcriptome data. For each patient, a ‘response’ signature was calculated as the mean expression of the genes where NAXIVA $\text{LFC} > 2$, and a ‘non-response’ signature where $\text{LFC} < -2$. Javelin¹⁶ patients with a ‘response’ signature in the top quartile and a ‘non-response’ signature in the bottom quartile were termed ‘NAX-RNA-hi’, and patients with ‘response’ in the bottom quartile and ‘non-response’ in the top quartile were termed ‘NAX-RNA-lo’. Median PFS was compared between the two groups, and a log-rank test was run using Survival (v3.7.0) and Survminer (v0.5.0) R packages.”

In Results, ***Responders and non-responders have distinct transcriptomic profiles***

“We generated signatures from the top differentially expressed genes in the NAXIVA patients. We applied these signatures to published patient level RNA-seq data from the Javelin Renal 101 study¹⁶, a randomised phase III clinical trial comparing axitinib plus avelumab with sunitinib in advanced renal cancer. We stratified the patients into NAX-RNA-hi and NAX-RNA-lo groups. Progression-free survival (PFS) was compared between patients in the NAX-RNA-hi and NAX-RNA-lo groups in each arm of the Javelin trial (Fig. 3d-e). For the sunitinib arm, the patients in the NAX-RNA-hi group had a higher PFS than those in the NAX-RNA-lo group (Fig. 3d). However, this effect was not true in the avelumab+axitinib arm of the trial (Fig. 3e).”

Figure 3

d-e, The most differentially expressed genes in NAXIVA ($p < 0.05$) stratify patients according to PFS in Javelin Renal 101 study for the Sunitinib arm (**d**) and not for the avelumab+axitinib arm (**e**).

Figure S11

a, The NAXIVA baseline signature can stratify patients in the sunitinib arm of the Javelin Renal 101 trial [15].

In Discussion, paragraph 4

“Our study finds that the expression of highly upregulated proteins in NAXIVA responders were also upregulated in C1/2 patients (angiogenic/angio-stromal) in publicly available IMmotion151 transcriptomic data, whereas the opposite was observed for C4/5/6 (T-effector/proliferative, proliferative and stromal/proliferative), which fitted better with the non-responding patients. We also find that the genes encoding our blood markers are able to stratify the patients in the Javelin Renal 101 trial according to progression-free survival.”

Comment 3

In Figure 1d and 1e, it would be beneficial to include a whole slide image as shown in Figure 1c. Furthermore, including several more representative images from different patients would provide more informative insights.

Response:

Thank you for this comment.

Changes implemented:

We have included further representative images, with whole slide scans and high magnification images from the VTT edge and core in a new figure (S2). We have also added a note to the methods to clarify that the whole slide scans and high magnification images were taken on different instruments.

In Methods, *Histology & Image Analysis*

“Whole slides were scanned at $\times 40$ magnification on the Zeiss Axio Scan Z1 system. High resolution images were acquired using a Leica SP5 Confocal Microscope at \$\times 40\$ objective magnification.”

Figure S2. Untreated microenvironment of VTT.
Additional scans of tumour and VTT.

a-b, Paired primary tumour and VTT cases stained for **(a)** vascular and **(b)** immune markers. CA9+ viable tumour fills the lumen of the renal vein in both cases.

c-d, Higher magnification confocal images of the VTT. **(c)** The VTT edge shows CD31+ endothelial cells and SMA+ stromal cells on the surface of the VTT (white arrows). The adjacent SMA + normal vein is visible. **(d)** Immune cells are visible in the VTT core.

Reviewer 3

Comment 1

Code not available.

Response:

We have submitted a Code Ocean capsule (27b34b3a-1ec6-4346-a487-ebe26113c0c9) and with title Angiogenic and Immune Predictors of Neoadjuvant Axitinib Response in Renal Cell Carcinoma with Venous Tumour Thrombus (18-551959-0).

Reviewer 4

Comment 1

A Figure/Table summarizing the study findings (i.e. all putative predictive biomarkers of response to neoadjuvant axitinib) in the form of a “graphical abstract” could serve readers who are less experienced in translational research and may enhance the impact of the study

Response:

Thank you for this idea. Our understanding is that graphical abstracts are not permitted in Nature Communications articles, but we have adjusted our Fig. 1a to include the impacts of the study.

Changes implemented:

We have added an outcomes box to our Fig. 1a.

Figure 1. Multiparametric investigation of VTT response in the NAXIVA trial.

a, Patients received up to 8 weeks of axitinib treatment. VTT response was evaluated by MRI at baseline, week 3 and week 9. Tissue was collected at baseline biopsy, and at surgery from the VTT and primary tumour. Serial blood samples were taken before, during and after treatment. Research samples were assessed by a range of techniques to identify markers of response. Baseline and week 3 parameters were combined in a machine learning model for treatment response.

Comment 2

Regarding the machine learning model integrating multiple baseline features predicts treatment response: the definition of response is not necessarily clinically meaningful (response is defined as a >30% reduction in VTT length compared to baseline). Could the

authors expand the definition on how such a definition of response translates into harder clinical endpoints in the study (any change in surgical approach, patient outcome or surgery-related variable?). For the ML model, could the authors evaluate other endpoints? (e.g. change in surgical strategy, reduction of surgery's invasiveness, RFS, etc.)?

Response:

Thank you for raising this important point.

We have used VTT length change as our primary endpoint because we believe it reflects the biological characteristics of the VTT lesions more completely than the Mayo classification. Relapse-free survival was not assessed in NAXIVA, and we believe that the choice of surgical approach is overly influenced by the preferences of the operating surgeons.

The Mayo classification is not linearly proportional to the change in VTT length: depending on the VTT location, some patients with smaller VTT length change are classified as responders, while some patients with larger VTT length change are classified as non-responders (Fig. S1b, reproduced below).

This poses a challenge, as we would not expect Mayo responders with a very small change in VTT length to share biological characteristics with those showing a large VTT length change. Although the Mayo classification is more clinically useful as it correlates with surgical approach, this non-linear relationship to VTT length change is challenging, and likely limiting, for identification of biological determinants of response.

In spite of these caveats, there is a strong correlation between VTT length and Mayo classification. 16 of the 20 patients fall into the same category by our VTT length-based classification. We find that the exact same model we used to predict VTT length, without retraining, is reasonably effective at predicting response as determined by Mayo, with AUC = 0.802 and 0.769 for the baseline and dynamic models, respectively.

Our focus on identifying the biological determinants of response led us to choose VTT length change over any other metric collected in the NAXIVA trial. We think that change of VTT length has a closer, more direct relationship with these biological features, and provides more insights into the mechanisms behind response. The model based on VTT length change effectively differentiates between patients and can guide surgical decisions due to its strong association with the Mayo classification.

Changes implemented:

We have produced an additional supplementary figure to summarise these results (Fig. S11d), and have added a comment in the results about the relevance of this model in surgical settings.

In Results, section ***Adding early dynamic measure improves the performance of the machine learning model*** (new text highlighted in blue)

“Interestingly, the week 3 fold-change in plasma sTie-2 and PIGF also showed potential for stratification. The baseline and dynamic models are capable of predicting surgically-relevant Mayo Classification-based response in these patients (Fig. S11d).”

Figure S11.
d, ML model prediction of response according to Mayo classification.

Comment 3

Assessing the specific analyses on the ML model to predict response is beyond this Reviewer's knowledge and should be extensively checked by an expert biostatistician.

Response:

Thank you for your comment.

Reviewer 5

Comment 1

Sample size: With only 20 patients, the study's findings are based on a small dataset, and the predictive power of the machine learning model could be compromised by this.

Response:

Thank you, and we agree that dataset size makes the modelling challenging. However, we have used a robust statistical approach (leave-one-out nested cross validation approach, with consensus-based feature importance) that maximises the training data available while ensuring that the model is tested on independent samples without any data leakage. These types of approach that rely on subdividing the dataset have the risk of under-powering models, resulting in no predictive power; but the fact that we find a powerful predictor with a small number of features most of which are very stable suggests that the signal is strong, and greatly reduces the chances of significant overfitting. (We address external validation in the next question.)

Changes implemented:

We have added a more detailed description of our leave-one-out cross validation approach in the discussion.

In Discussion, paragraph 8 (new text highlighted in blue)

“The models are limited by a small dataset, but our approach (leave-one-out nested cross-validation with consensus-based feature importance) effectively maximizes training data and tests the model on independent samples. The discovery of a robust predictor with a limited and highly stable set of features indicates a strong signal and substantially reduces the risk of overfitting.”

Comment 2

Absence of external validation: The model's generalizability would be improved with validation on an independent cohort.

Response:

Thank you for making this important point about validation.

We agree that it would be ideal to have an external validation dataset; however, we are not aware of any other clinical trials or studies with exactly equivalent data. Future validation work will involve an ongoing clinical trial led by our team, WIRE (NCT03741426), a novel multi-centre trial which will measure the same markers as tested in NAXIVA. We anticipate that WIRE will serve as future validation for the role of CCL17, IL-12p70, and the dynamic markers PIGF and sTie-2. However, WIRE is expected to continue to recruit patients for several years, so it is out of scope for this paper.

In response to the reviewer's comments, we have conducted an in-depth analysis of existing trials which administer an anti-angiogenic / TKI monotherapy and have publicly available molecular data that is similar enough to NAXIVA that a proxy signature can be built.

The trials we have considered for external validation are in the table below.

Trial	Feature type	Primary endpoint	Drug	Patient population	PMID
JAVELIN Renal 101 (NCT02684006)	Transcriptomics	Progression-free survival	Sunitinib, Avelumab + Axitinib	Advanced RCC	32895571 ¹
IMmotion151 (NCT02420821)	Transcriptomics	Progression-free survival NB: PFS data not published	Sunitinib, Atezolizumab + Bevacizumab	Advanced RCC	33157048 ²

These trials have transcriptomic instead of protein data, so in order to validate our signature we have identified the genes that correspond to our predictive plasma proteins, and used them as a proxy for the NAXIVA signature.

First, using the IMmotion151 trial, we validated the main ‘biological’ finding of our paper, namely that NAXIVA responders have an ‘angio-high’ profile, whereas non-responders have an ‘immuno-high’ profile. We find that the genes that characterise NAXIVA responders belong to IMmotion clusters 1 or 2 (anti-angiogenic therapy-responsive clusters), while the genes that characterise NAXIVA non-responders are overrepresented in IMmotion cluster 4 (immunotherapy-responsive). These results nicely confirm our NAXIVA findings.

Next, we built a NAXIVA signature-proxy by averaging out the main expression levels of the relevant genes. We then stratified JAVELIN Renal 101 patients according to the NAXIVA signature-proxy, assigning ‘Nax-hi’ to patients in the top 25th percentile, and reverse for ‘Nax-lo’. There was a significant survival difference between the two groups in the sunitinib (i.e. anti-angiogenic therapy) arm of the trial, supporting the validity of the features identified by our baseline model.

Changes implemented:

We have included these graphs in the main figures and supplementary figures, and edited our text to explain the methods for external validation.

In Methods, ***Machine learning models***

“External validation: No other studies have published the blood plasma markers and histopathological features following treatment with anti-angiogenic therapy in ccRCC with VTT. However, we leveraged publicly available transcriptome datasets from Phase III trials in advanced ccRCC to validate our findings.

(1) We assigned each patient in the Javelin Renal 101 transcriptomics set¹⁵ a ‘response’ and ‘non-response’ pseudo-signature score using the mean expression of the genes encoding the proteins identified in the NAXIVA baseline model. For response, the genes used were *PECAM1* (encoding CD31) and *CD34* (encoding CD34). The non-response genes used were *CCL17*, *CCR4* (the *CCL17* receptor, which our scRNA-seq data suggests is expressed by CD4+ T cells in tumour tissue, Fig. S10a), *IL12A* and *IL12B* (the genes encoding the IL-12p70 subunits). The patients in the top quartile for response and in the bottom quartile for non-response were included in the ‘NAX-hi’ group (n=47). The patients in the bottom quartile for response and in the top quartile for non-response were included in the ‘NAX-lo’ group (n=28). Survival analyses of these two groups was done using a log rank test using the Survival (v3.7.0) and Survminer (v0.5.0) R packages.

(2) The genes in both signatures were mapped onto the differential expression data from the IMmotion151 transcriptomic dataset¹⁷.”

In Results, ***A machine learning model integrating multiple baseline features predicts treatment response***

“The expression of the genes encoding these baseline features (*PECAM1*, *CD34*, *CCL17*, *CCR4*, *IL12A*, *IL12B*) identified by the model are able to stratify the patients in Javelin Renal 101¹⁵ according to progression-free survival (log-rank test, p = 0.034, Fig. S11a).”

Figure S11

a, The NAXIVA baseline signature can stratify patients in the sunitinib arm of the Javelin Renal 101 trial [15].

In Discussion, paragraph 4

“Our study finds that the expression of highly upregulated proteins in NAXIVA responders were also upregulated in C1/2 patients (angiogenic/angio-stromal) in publicly available IMmotion151 transcriptomic data, whereas the opposite was observed for C4/5/6 (T-effector/proliferative, proliferative and stromal/proliferative), which fitted better with the non-responding patients. We also find that the genes encoding our blood markers are able to stratify the patients in the Javelin Renal 101 trial according to progression-free survival.”

Comment 3

Lack of functional validation: The study relies on correlative biomarkers without experimental validation of their causal role in response to treatment. These limitations do not prohibit publication but should be addressed in future studies.

Response:

This is an important comment. Validating the functional effect of each of these markers is beyond the scope of the study, but we acknowledge that a further understanding of the biological mechanism by which each of these markers may be exerting their effects is important for further development of our proposal to improve treatment decision-making in ccRCC with VTT.

Changes implemented:

We have made an additional comment on this in the discussion.

In Discussion, final paragraph

“Our investigations of the microenvironment and blood features have identified predictive biomarkers that might be clinically and functionally validated in these studies, either alone or as a combined assay.”

References

1. Motzer, R. J. *et al.* Avelumab plus axitinib versus sunitinib in advanced renal cell carcinoma: biomarker analysis of the phase 3 JAVELIN Renal 101 trial. *Nat. Med.* **26**, 1733–1741 (2020).
2. Motzer, R. J. *et al.* Molecular Subsets in Renal Cancer Determine Outcome to Checkpoint and Angiogenesis Blockade. *Cancer Cell* **38**, 803-817.e4 (2020).
3. Reese, A. C., Whitson, J. M. & Meng, M. V. Natural history of untreated renal cell carcinoma with venous tumor thrombus. *Urol. Oncol. Semin. Orig. Investig.* **31**, 1305–1309 (2013).
4. Martínez-Salamanca, J. I. *et al.* Lessons learned from the International Renal Cell Carcinoma-Venous Thrombus Consortium (IRCC-VTC). *Curr. Urol. Rep.* **15**, 404 (2014).